

# Mass evolution of the Antarctic Peninsula over the last two decades from a joint Bayesian inversion

Stephen J. Chuter[1], Andrew Zammit-Mangion[2], Jonathan Rougier[3], Geoffrey Dawson[1], Jonathan L. Bamber[1]

[1] Bristol Glaciology Centre, School of Geographical Sciences, University of Bristol, Bristol, UK
[2] School of Mathematics and Applied Statistics, University of Wollongong, Wollongong, New South Wales, Australia
[3] Rougier Consulting Limited

*Correspondence to*: Stephen J. Chuter (s.chuter@bristol.ac.uk)

**Abstract.**

The Antarctic Peninsula has been an increasingly significant contributor to Antarctic Ice Sheet mass losses over the last two decades. However, due to the challenges presented by the topography and geometry of the region, there remain large variations in mass balance estimates from conventional approaches and in assessing the relative contribution of individual ice sheet processes. Here, we use a regionally optimised Bayesian Hierarchical Model joint inversion approach, that combines data from multiple altimetry studies (ENVISAT, ICESat-1, CryoSat-2 swath), gravimetry (GRACE and GRACE-FO) and localised DEM differencing observations, to solve for annual mass trends and their attribution to individual driving processes for the period 2003-2019. The region experienced a mass imbalance rate of -19±1.1 Gt yr$^{-1}$ between 2003 and 2019, predominantly driven by accelerations in ice dynamic mass losses in the first decade and sustained thereafter. Inter-annual variability is driven by surface processes, particularly in 2016 due to increased precipitation driven by an extreme El Niño, which temporarily returned the sector back to a state of positive mass balance. In the West Palmer Land and the English Coast regions, surface processes are a greater contributor to mass loss than ice dynamics in the early part of the 2010s, although both processes are acting simultaneously. Our results show good agreement with conventional and other combination approaches, improving confidence in the robustness of mass trend estimates, and in turn, understanding of the region's response to changes in external forcing.

## 1. Introduction

Over the last two decades, the Antarctic Peninsula (AP) and Bellingshausen Sea Sector of West Antarctica have experienced increased mass loss (Wouters et al., 2015; Martín-Español et al., 2016; McMillan et al., 2014; Sutterley et al., 2014), grounding line migration (Christie et al., 2016) and increases in ice sheet velocity (Mouginot et al., 2014; Gardner et al., 2018). The collapse of the Larsen A and Larsen B ice shelves in 1995 and 2002, respectively, and the subsequent reduction in buttressing they provide to the grounded ice sheet, resulted in rapid thinning (Scambos et al., 2014; Rott et al., 2018) and increased glacier flow velocities (Scambos et al., 2004) in the northern AP. These events are seen as potential analogues for a future response of the West Antarctic Ice Sheet (WAIS) to further ice shelf thinning or collapse, particularly over areas susceptible to marine ice sheet instability (Schoof, 2007). It is therefore important to have continuous long-term monitoring of the AP, as it can help



us better understand the multi-decadal ice sheet response to external oceanic and atmospheric forcing, subsequently improving process representation in ice sheet models. Additionally, the AP has accounted for an increasingly significant component of

the ice sheet's contribution to global sea level rise over recent decades (Bamber et al., 2018; Global Sea Level Budget Group, 2018; Shepherd et al., 2018). Therefore, robust estimates of its mass changes are imperative to provide sea-level budget closure. It is a region, however, that has proved difficult to monitor using conventional mass balance approaches because of the high relief and northerly latitude. The former has resulted in relatively poor coverage by satellite altimetry, Interferometric SAR-derived (InSAR) velocities and ice thickness data, as well as providing significant challenges for regional climate models due

to the large climatic gradient across the AP. The latter means that the across-track spacing of satellite altimetry is significantly poorer than parts of Antarctica that lie further south.

These coverage issues have made it difficult to reconcile mass balance estimates for the AP from conventional approaches, with altimetry and mass budget techniques not agreeing within error for the period 2003-2010 (Shepherd et al., 2018). At the

individual basin scale, even estimates from the conventional approaches using the same observations do not agree within error for common time periods (Wouters et al., 2015; McMillan et al., 2014). This suggests uncertainties and biases in the methodological assumptions and model corrections are being propagated into the resulting mass balance estimates. By contrast, data-driven approaches which can simultaneously assimilate the diverse observations available allow us to exploit their individual advantages, whilst reducing the sources of uncertainty and assumptions required by single technique approaches

(Martin-Espanol et al, 2016). Robustly combining these observations, however, is challenging due to their markedly different spatio-temporal characteristics and sources of common error.

To address this issue, we apply a Bayesian Hierarchical Model (BHM) framework (Zammit-Mangion et al., 2014, 2015a, b) to resolve for the spatio-temporal mass trends and component parts for the AP from 2003 to 2019; an approach which has been

previously demonstrated at the ice sheet scale for a shorter time period (Martín-Español et al., 2016). We develop and optimise the BHM framework for this high-relief region of Antarctica by both improving the spatial resolution and including new observation datasets. An advantage of the BHM approach is its ability to incorporate irregular data in both space and time, while explicitly accounting for error propagation in the results. This means we can exploit localised datasets such as elevation change from high resolution Digital Elevation Model differencing (dDEM), as well as addressing the gap in the gravimetry

record between the decommissioning of GRACE in 2017 and the launch of GRACE-FO in 2018. The combination of a diverse range of observations enables us to constrain the mass balance of the region with high confidence and provide insights into the driving processes. We use these mass trend estimates and their component parts to explore the discrepancies between different approaches for the West Palmer Land region in the southern AP, one of the most challenging regions of the ice sheet to observe and the source of greatest divergence in previous mass balance estimates.





## 2. Data


For the 2003 to 2019 period of this study, observations of ice sheet elevation and mass change from multiple missions are used as inputs to the BHM framework. The spatial domain of our study is shown in Figure 1a, with altimetry dataset coverage shown in Figure 1b and region specific datasets for the northern AP in Figure 1c. An overview of datasets in the framework is additionally given in Table 1. As our framework solves for the rate of elevation change in each ice sheet process at a calendar

year resolution, our datasets represent the rate of change in either elevation or mass. All datasets are processed to represent calendar year trends, with differing methodological approaches given the variations in the spatio-temporal sampling of each dataset. The subsections below focus on the processing methodologies for each category of observation.

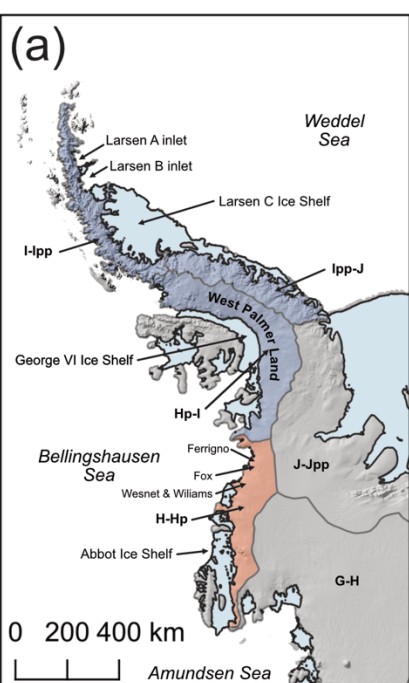

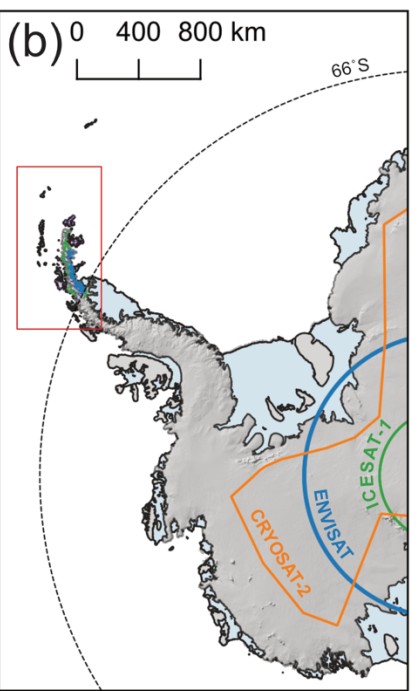

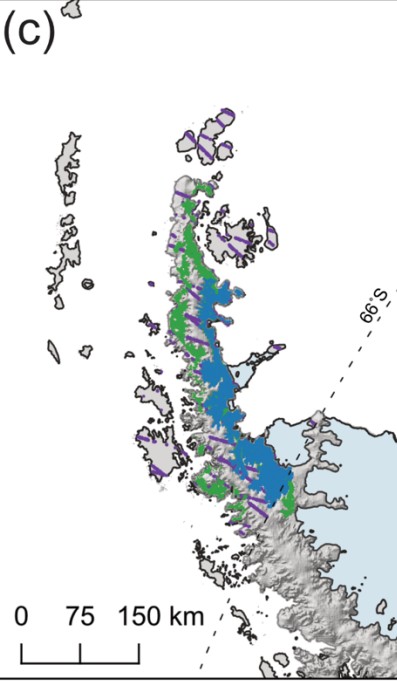

**Figure 1- (a)** Region of interest for this study, overlain with drainage basin definitions (Rignot et al., 2011b, 2013). Basins included in the Ice Sheet Mass Balance Intercomparison Exercise (IMBIE) (Shepherd et al., 2018) definition of the AP are shaded in purple, with orange basins indicating those outside the AP but included in this study. **(b)** Spatial coverage of altimetry and DEM datasets for the region. The latitudinal limit of ICESat-1 and ENVISAT are shown in green and blue, respectively. Orange line indicates the CryoSat-2 POCA/Swath to Low Resolution Mode (LRM) transition (with LRM coverage south of the orange delineation). The dashed latitudinal line at 66°S indicates

the separation between ICESat-1 data used over the interior of the ice sheet (Martín-Español et al., 2016) and the optimised data over the northern AP (Scambos et al., 2014). **(c)** Zoomed spatial coverage of the red bounding box in panel (b), indicating the datasets used over the northern AP. Purple and green regions indicate coverage of the optimised ICESat-1 data and dDEM data, respectively, from Scambos et al. (2014). Blue regions indicate coverage of dDEM data from Rott et al. (2018). All data is overlain on the hill shade of the Reference Elevation Model of Antarctica REMA DEM (Howat et al., 2019).




**Table 1 -** Summary of observations and prior auxiliary datasets used in this study, including their spatio-temporal coverage and latent processes they observe.

| Observation/Model | Role in BHM | Processes | Temporal Coverage | Spatial Coverage | Reference |
|---|---|---|---|---|---|
| GRACE | Observation | GIA, SMB, Ice Dynamics | 2003-2016 & 2019 (annual) | 0.5° x 0.5° | (Wiese et al., 2018; Watkins et al., 2015; Wiese et al., 2016; Landerer et al., 2020) |
| ENVISAT | Observation | GIA, SMB, Firn, Ice Dynamics | 2003-2009 (annual) | 5 km | (Flament and Rémy, 2012) |
| ICESAT northern AP (<66°S) | Observation | GIA, SMB, Firn, Ice Dynamics | 2003-2009 (annual & multi-annual) | 1km | (Scambos et al., 2014) |
| ICESAT (>66°S) | Observation | GIA, SMB, Firn, Ice Dynamics | 2003-2009 (annual) | 10km | (Martín-Español et al., 2016) |
| CryoSat-2 Swath (CryoTempo-EOLIS) | Observation | GIA, SMB, Firn, Ice Dynamics | 2011-2019 (annual) | 10 km (regions covered by Cryosat-2 SARIn mode) | (Gourmelen et al., 2018) |
| TanDEM-X | Observation | GIA, SMB, Firn, Ice Dynamics | 2011-2016 (multi-annual) | 1km | (Rott et al., 2018) |
| ASTER/SPOT-5 | Observation | GIA, SMB, Firn, Ice Dynamics | 2001-2010 | 1km | (Scambos et al., 2014) |
| GPS | Observation | GIA | 2003-2009 | N/A | (Martín-Español et al., 2016) |
| RACMO 5.5 km | Prior (For AP Sector) | SMB | 2003-2016 | ~ 5.5km | (Van Wessem et al., 2016) |
| RACMO2.3p2 27km | Prior (Other Sectors) | SMB | 2003-2016 | ~ 27km | (Van Wessem et al., 2014) |
| MEaSUREs Ice Sheet Velocity Mosaic (Version 2) | Prior | Ice Dynamics | 1996-2016 | 450m | (Rignot et al., 2017) |
| Vertical crustal displacement due to GIA | Prior | GIA | N/A | 20km | (Martín-Español et al., 2016) |

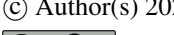



## 2.1 Gravimetry

In this study we use the NASA Jet Propulsion Laboratory (JPL) RL06_v02 global monthly mascon product to provide observations of mass change from for the GRACE and GRACE-FO mission for the time period 2003 to 2019 (Wiese et al., 2018; Watkins et al., 2015; Wiese et al., 2016; Landerer et al., 2020). The dataset consists of monthly mass change anomalies with respect to a 2004-2009 time-mean baseline, posted on a 0.5° x 0.5° degree global grid. We use the product which has not been processed using the Coastline Resolution Improvement (CRI) filter, as the BHM accounts for partitioning of signals

within an areal observation using other available observations such as altimetry. Therefore, the inclusion of a CRI filtered product could introduce inconsistencies between observations that are a result of dataset processing choices as opposed to differences in the actual geophysical signals that are being observed. The ICE6G-D (Peltier et al., 2018) GIA model correction is also restored to the observations prior to trend determination, as we explicitly account for mass changes due to GIA as a process in our framework (see methods).


Whilst the observations are provided at a 0.5° x 0.5° degree grid resolution, equal area 3° x 3° spherical cap mass concentrations were used to estimate the gravity fields in the JPL processing, and this determines the native resolution of the product. Our framework benefits from the assumption of uncorrelated uncertainties in each observation dataset (Martín-Español et al., 2016). Therefore, the high spatial resolution mascons were aggregated, by combining any neighbouring mascons with the same

observed values, so that the measurement-error in this product can be assumed to be spatially uncorrelated. For each aggregated mascon time series, annual trends were determined through the fit of a piecewise linear trend to the de-seasoned time series, with knots at January 1$^{st}$ of each calendar year to ensure consistency at the year boundary. Uncertainties on the observed trends were calculated as the regression error of the piecewise fit. Due to the temporal gap between degradation and decommissioning of the GRACE satellite in 2017 and the launch of GRACE-FO in May 2018, trends for the calendar years 2017 and 2018 were

not estimated.

## 2.2 Altimetry

### 2.2.1 ENVISAT

Surface elevation trends from ENVISAT are utilised in the framework for the period 2003-2010, providing complementary coverage to ICESat-1 trends over the same period. ENVISAT provides higher spatial density observations at lower latitudes,

but with reduced accuracy in regions of ice sheet gradients, such as near the grounding line. Elevation trends follow the processing used in the previous Antarctic wide model implementation (Flament and Rémy, 2012; Martín-Español et al., 2016), where trends are estimated along the satellite ground track using a 10 parameter plane fitting algorithm at 1 km intervals.



Annual trends were determined using a 3-year moving window on the time series and are then subsequently aggregated into 20 km grid cell spacings.

### 2.2.2 ICESat-1

We use two ICESat-1 $\Delta h/\Delta t$ products, for the period 2003–2009. One processing chain is optimised for the mountainous geography of the northern AP (<66°S) (Scambos et al., 2014), while the other processing chain used in previous implementations of the BHM framework for the Antarctic continent (Martín-Español et al., 2016) is optimised for ice sheets with lower surface slopes (Fig 1b).

For regions >66°S ICESat-1 release data 633 data were used, following the same processing methodology as the previous BHM implementation (Martín-Español et al., 2016). These data are corrected for the Centroid versus Gaussian offset (Borsa et al., 2014) and inter-campaign biases (ICB) (Hofton et al., 2013). An along-track plane regression approach (Howat et al., 2008; Moholdt et al., 2010) was used to determine annual elevation change rates ($\Delta h/\Delta t$) from a 3-year monthly rolling window. After determination of elevation rates potential outliers with a plane-fit uncertainty of >2 m yr$^{-1}$ were removed. This is larger than the threshold used in Martín-Español et al (2016b), but was selected to increase the data availability over regions of more heterogenous terrain such as the southern AP and the Abbot region of West Antarctica. The elevation rates and uncertainties were averaged on a 20 km grid (Polar Stereographic Projection – EPSG:3031).

For the northern AP (<66°S) we use ICESat-1 elevation trends from Scambos et al (2014), which uses a regional high resolution DEM to perform outlier filtering and slope correction for each campaign to the reference track (Cook et al., 2012). Only elevation rates from campaign combinations that are near whole year separation were used (to reduce potential seasonality effects in the calculation of $\Delta h/\Delta t$), in addition to any campaign combination with a time separation greater than 4 years being removed. Since the BHM calculates mass trends at annual (calendar year) time steps, an annual trend is only assigned if the ICESat-1 trends cover more than 75% of any given calendar year, to reduce the influence of temporal seasonality being erroneously incorporated into the linear trend signal. Uncertainties of 0.15 m yr$^{-1}$ and 0.30 m yr$^{-1}$ were used for elevation rates calculated > 1000 m.a.s.l and < 1000 m.a.s.l with respect to the Cook et al (2012) DEM, respectively (Scambos et al., 2014).

The original along-track 43.75 m postings (Scambos et al., 2014) of $\Delta h/\Delta t$ rates were downscaled through averaging onto a quasi-regular grid at 1 km polar stereographic grid (EPSG:3031), separately for each campaign combination. Uncertainties of the lower resolution $\Delta h/\Delta t$ rates were calculated through error propagation of the original uncertainty with the standard deviation of elevation rates within each downscaled grid cell. For $\Delta h/\Delta t$ rates calculated from repeat tracks separated by multiple years, we inflate the uncertainties as follows (Martín-Español et al., 2016)

$$\sigma\sqrt{n} \tag{1}$$



Where $\sigma$ is the measurement uncertainty and $n$ is the number of years the observations represent. These trends and inflated errors are then used as the observation for each year that the trend covers in the framework (e.g. assuming no variation in elevation rate over the timespan of trend calculation). The error inflation has the practical impact of reducing the influence of observations that represent multiple years in the BHM, as whilst they provide useful information on the ice sheet state they will not be as informative for short temporal length scale processes as observations derived over shorter temporal baselines.

### 2.2.3 CryoSat-2


CryoSat-2 has provided high resolution observations of ice sheet elevation change since 2010, with its SARIn mode of operation around the ice sheet periphery increasing coverage and accuracy compared to conventional radar altimetry. Recent developments in Swath processing of the SARIn waveforms has resulted in orders of magnitude more data coverage around these key ice sheet regions (Gray et al., 2013; Gourmelen et al., 2017a), increasing the temporal and spatial resolution at which

surface elevation changes can be determined.

In this study we use the ESA CryoSat-2 CryoTempo-EOLIS swath point observation data product from September 2010 to December 2019 (Gourmelen et al., 2017b). All swath observations that met the quality criteria of the CryoTempo-EOLIS product are utilised. Rates of elevation change were determined through a plane fitting approach at regular grid spacing across

a polar stereographic grid (McMillan et al., 2014), using the following linear model within each grid cell:

$$z(x, y, t) = \bar{z} + a_1 x + a_2 y + a_3 xy + a_4 t + \varepsilon \qquad (2)$$

where $z$ is the surface elevation, $\bar{z}$ is the mean surface elevation at the grid cell centre $t_0$ (temporal midpoint of fit), $x$ and y

are the spatial co-ordinates (Polar Stereographic projection EPSG:3031), $t$ is time and $\varepsilon$ is the observation noise. A grid resolution of 5 km is used, with annual trends calculated using a 3 year moving window centred on the year midpoint ($t_0$) for which the trend is determined. To eliminate potentially poorly fitted models, grid cells were removed from further analysis if any of the following conditions were met: an absolute $\Delta h/\Delta t > 15$ m yr$^{-1}$, surface slopes $> 3°$ or an elevation rate $\sigma > 1$ m yr$^{-1}$. In addition, a Median Absolute Deviation (MAD) kernel was used to remove outlier $\Delta h/\Delta t$ values $> 2$ MAD with respect to

neighbouring cells.

### 2.3. dDEM $\Delta h/\Delta t$

High spatial resolution $\Delta h/\Delta t$ observations derived from stereo-photogrammetry and DEM differencing (dDEM) (Fig. 1c) are available over the northern AP (Scambos et al., 2014; Rott et al., 2018) and provide the capability to measure localised changes in glacier elevation in mountain valley regions which are typically poorly resolved by satellite altimetry. The ability of the

BHM approach to assimilate spatially irregular data means that, for the first time, these small-scale local observations can be



used as part of a regional assessment. We used two different dDEM Δh/Δt datasets in this study (Rott et al., 2018; Scambos et al., 2014), derived from Advanced Spaceborne Thermal Emission and Reflectance Radiometer (ASTER), Satellite Pour l'Obervation de la Terre 5 (SPOT-5) and TanDem-X observations.

For the 2003-2010 period, elevation trends from a combination of 16 ASTER and SPOT-5 images from Scambos et al (2014) are used. The data product provides multi-annual elevation trends at 50 m postings for the northern AP (<66°S) region, with an uncertainty in elevation rate of 0.3 m yr$^{-1}$ when compared with ICESat-1 elevation trends at co-incident locations (Scambos et al., 2014). As an initial step, all Δh/Δt estimates calculated in regions >1000 m.a.s.l. were removed from the dataset, due to the lack of high-contrast features in heterogenous topographic regions causing image correlation issues (Scambos et al., 2014).

We then aggregated all remaining data to a 1 km resolution through averaging of the Δh/Δt rates and calculation of the Mean Absolute Error (MAE) for each new low-resolution grid cell (Pogson and Smith, 2015). We then propagated the uncertainties of the original resolution observations with the MAE values of the down sampled 1 km grid cells to attain representative uncertainties for each aggregated Δh/Δt value. As the trends represent the multi-annual linear rate of change between two SAR acquisitions, we inflate the trend uncertainty using the approach of Eq.(1).


Ice sheet Δh/Δt trends from Rott et al (2018), calculated through differencing of repeat TanDEM-X acquisitions, were used for the 2011-2016 period. Δh/Δt observations were available for the 2011-2013 period over the Larsen A region, with two sets of observations (2011-2013 & 2013-2016) covering the Larsen B region of the northern AP at a spatial resolution of 12 m². The advance of the coastline between 2013-2016 over the Hektoria, Green and Crane glaciers has resulted in the presence of a large

ice thickening trend in these observations at the outlets of these glaciers. As these features are not situated on grounded ice, therefore playing no role in the mass balance of the ice sheet, they are removed from the dataset using a threshold of > +5 m yr$^{-1}$ elevation change. This ensures this thickening pattern was not included in the subsequent down sampling and aggregation processing for pixel at the grounding line boundary.

To use the TanDEM-X dataset in the framework, observations were resampled to a 1 km polar stereographic grid (EPSG:3031) by calculating the mean Δh/Δt and MAE per grid cell. The uncertainties on these down sampled observations were calculated through error propagation of the uncertainties of the original observations (Rott et al., 2018) with the MAE of each grid cell (Pogson and Smith, 2015). Any down sampled values within 5 km of the grounding line (Depoorter et al., 2013) were subsequently removed to ensure no values at the ice sheet margin, where grounding line migration could be present, were

included in the framework. Finally, due to these Δh/Δt observations being representative of multi-annual trends, the observation uncertainties were scaled in magnitude by the length of their temporal coverage in accordance with Eq (1).

**2.4 GPS**



We use elastically corrected GPS observations of vertical land motion for the period 2003-2009 from Martín-Español et al.
(2016) as an observation constraint for the temporally-invariant GIA process in the framework. Details of the processing,
including trend estimation and elastic correction processing approach is outlined in Martín-Español et al. (2016). It is important
to note that we are not solving for GIA in this study, but instead using the Martín-Español et al. (2016) spatial solution as a
hard constraint on the process (see section 3.2). As these GPS observations are the same as those used to produce the data-
driven GIA solution, they act to provide an extra constraint on this process in the framework.

## 3. Methods

A BHM approach has been developed for use over the whole Antarctic Ice Sheet for determining both time-invariant mass
balance (Zammit-Mangion et al., 2014, 2015a, b; Schoen et al., 2015) and time-evolving mass balance (Martín-Español et al.,
2016) of the constituent driving processes. Here, we adapt the framework used in Martín-Español et al. (2016), by addressing
the limitations of that study in determining mass trends in the AP. The main limitation we address is the model of temporal
evolution of the ice dynamics process that was used. In Martín-Español et al. (2016), a simple linear or quadratic trend was
used, which is less justifiable in this case due to the longer time period we consider. We also refine the mesh resolution for
each ice sheet process over the AP, and use prior distributions that are more representative and derived from products optimised
for the region (e.g. the RACMO 5.5 km Surface Mass Balance (SMB) model). In the following subsections we outline the
conceptual overview of the framework and these methodological developments.

### 3.1 Finite Element Meshes

The smaller geographical area (Fig. Figure 1) we consider, when compared to the previous time-evolving continental scale
study (Martín-Español et al., 2016), allows for higher density Finite Element Meshes (FEMs) in regions where there is likely
to be higher spatial heterogeneity in ice sheet processes (e.g., small topographically constrained outlet glaciers in the AP). The
internal boundary of each process mesh is prescribed from the grounding line and island extents of Rignot et al. (2013) and an
external boundary of 600 km outwards from the grounding line. The lengths of the edges in the ice mesh range from 5 km to
100 km, with higher mesh density near the grounding line and in regions of ice flow velocity > 200 m yr$^{-1}$ (Rignot et al., 2017,
2011c; Mouginot et al., 2012). The surface process mesh has a minimum and maximum vertex length of 20 km and 100 km,
respectively, with higher resolution closer to the grounding line. The glacial-isostatic adjustment (GIA) mesh is constructed
without the internal grounding line boundary as it is a process that is present across the whole spatial domain, with a minimum
mesh resolution of 20 km.

### 3.2 Bayesian Hierarchical Model (BHM)

The BHM framework consists of three layers: the observation layer, the latent physical process layer, and the parameter layer,
which contains prior information about unknown parameters (Fig. 2). The observation layer establishes the links between the



observations used in the framework and the ice sheet processes they observe. The latent physical process layer describes how each ice sheet process evolves both spatially and temporally across the FEMs, represented as elevation trends per process for each calendar year time step. The prior layer provides information on spatio-temporal wavelengths of each ice sheet process, which is derived from analysis of forward geophysical models and auxiliary datasets. Whilst this means the approach is not completely independent of forward geophysical models, only the spatio-temporal properties of the forward models (e.g., spatial

and temporal length scales, which are broadly accurately represented) are used to aid source separation and not the specific outputs, as in the case of conventional approaches to mass balance. Therefore, the approach is largely independent of potential biases and uncertainties in the forward models, which are, in general, difficult to determine.

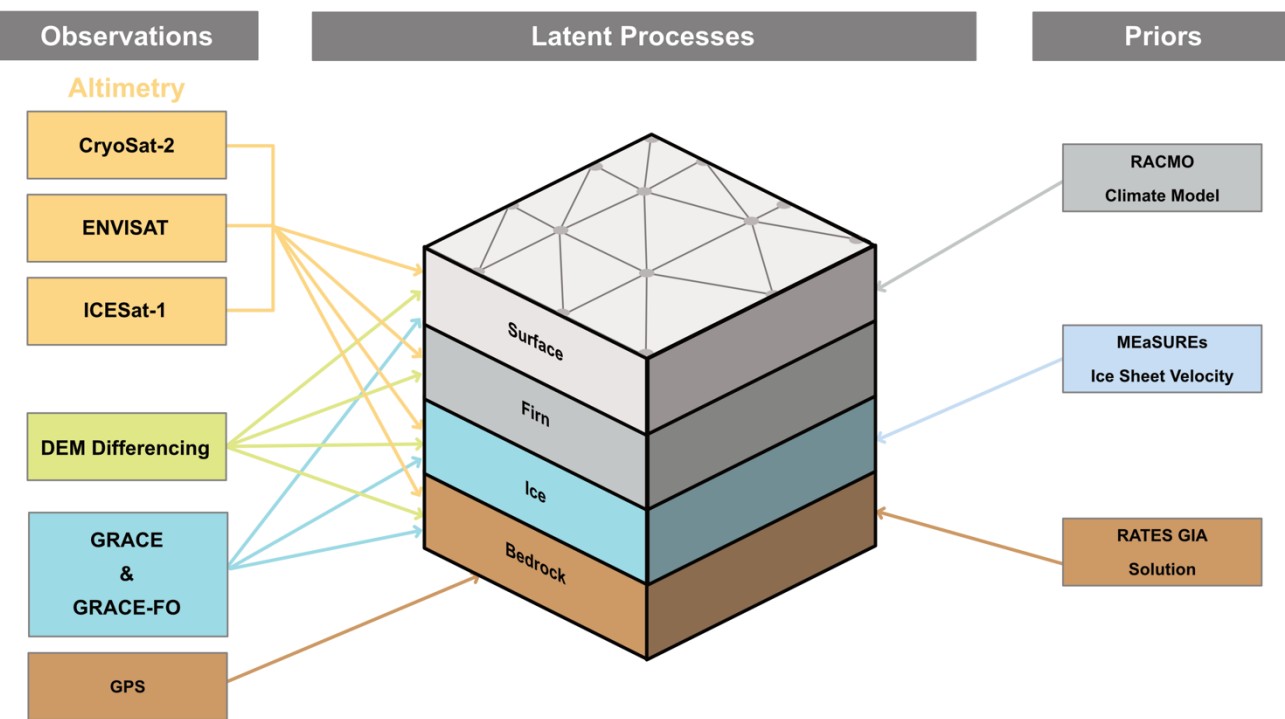

**Figure 2 -** Conceptual overview of the BHM framework. The left-hand side shows the observations used in the framework, with arrows linking the observations to the latent processes they observe. The centre panel outlines the ice sheet processes being solved for in the BHM using the FEM meshes (grey triangles and circular mesh nodes). The right-hand panel shows the prior information used to aid source separation of the observations into the latent process**s.**

The large scale ice sheet solution of Martín-Español (2016) produced a data-driven temporally-invariant estimate of vertical

land uplift rates due to GIA. We use this GIA solution as an informative prior (i.e. as the prior mean field) on the GIA latent process in this model, additionally constrained by the GPS observations in Martín-Español (2016). This means we are not solving for a new GIA solution, but instead enforcing a hard constraint on the latent process and follows the iterative solution approach of Martín-Español (2016). The velocity field used to help inform the spatial variance in the ice dynamics process



(Schoen et al., 2015) has also been updated to the MEaSUREs V2 composite velocity product (Rignot et al., 2017, 2011c;
Mouginot et al., 2012). The coverage of 99.6% in the updated product (Mouginot et al., 2017) precludes the need for balance
velocities to be used fill missing data regions. This is particularly beneficial in the mountainous interior of the southern AP,
where coverage from InSAR data was previously limited.

To determine the spatio-temporal length scales of surface mass balance over the mountainous terrain in the AP, we use the
RACMO2.3p2 5.5 km model. Its higher resolution improves its capability to resolve large SMB gradients from west to east
over the northern AP, in addition to the narrow outlet glaciers feeding into the ice shelves, when compared to the ice sheet
scale ~27 km resolution model (van Wessem et al., 2016). We use RACMO 5.5 km SMB anomalies w.r.t 1979-2002 baseline
for drainage sectors incorporated in the AP (Fig. 1) as outlined in the most recent Ice Sheet Mass Balance Inter-comparison
Exercise (IMBIE) assessment (Shepherd et al., 2018).  For sectors outside the AP, the ice sheet wide RACMO 27 km 2.3p2
model is used. As only spatio-temporal spectral properties are determined from the Regional Climate Models (RCMs), the use
of models at two different resolutions does not bias or create discontinuity in our prior for the surface latent process field. This
is an advantage of our approach: by not using the model output per se, we are able to incorporate prior information for each
region without introducing discontinuities.

Recently, satellite remote sensing observations of ice sheet behaviour have shown complex spatio-temporal evolution patterns
in ice dynamics, such as Pine Island Glacier (Bamber and Dawson, 2020). As a result, the linear and quadratic functions used
to describe the temporal evolution of ice dynamics in previous studies using this framework (Martín-Español et al., 2016) may
not adequately represent the complexity of the dynamic signal over longer (multi-decadal) time series. In the BHM we therefore
allow for a less restrictive temporal evolution of mass change due to ice dynamics. Specifically, we use a first order
autoregressive (AR(1)) process to describe the temporal evolution of ice dynamics. We fix the autoregression coefficient to
0.8, which implies a strong temporal dependence in time and consistent with marine ice sheet models operating at the same
temporal resolution (Robel et al., 2018). Model analysis of response times of glaciers draining into West Antarctica to
instantaneous forcing range from 6.4 years (Ferrigno Ice Stream) to 18.3 years (Thwaites) (Williams et al., 2012), equating to
AR(1) coefficients of 0.73 – 0.90 (Storch and Zwiers, 1999). Therefore, the selection of 0.8 for the AR(1) coefficient provides
an appropriate prior description of the smoothness of the process, whilst allowing for sufficient flexibility to capture changes
that may occur in response to variations in ocean forcing. This new latent process behaviour also allows the framework far
greater flexibility over longer time scales, future-proofing the approach for multi-decadal studies.

## 4. Results

Our results indicate that over the 17-year period of our study (2003-2019), the AP has experienced a mean mass imbalance of
-19.1± 2.0 Gt yr$^{-1}$, with the largest rates of mass changes occurring between 2008 and 2015 (Fig. Figure 3 & Table 2). In the





last three years of our study, reductions in mass imbalance are seen across all basins primarily because of an extreme snowfall event in 2016. The underlying driver of region-wide mass imbalance is dynamic mass losses across the sector (-19±1.1 Gt yr⁻¹), with no emergent trend in the surface processes present over the study period. As can be seen from Figure 3 however, the large inter-annual variability in mass balance is a result of the superposition of the SMB changes in the total signal, which are large enough to temporarily change the mass balance state of the whole sector for an individual year (e.g. 2016 and 2017). Therefore, the use of time average mean mass balance estimates over short temporal baselines for this sector can be misleading.

**Table 2 –** Mean mass trends for the 2003-2019 period of the study and over various epochs for the AP region and individual basins (see Fig. 1 for drainage basin and region definitions (Rignot et al., 2013, 2011a)) All uncertainties states as 1 σ.

|  | 2003 – 2019 (Gt yr⁻¹) | 2003 – 2007 (Gt yr⁻¹) | 2008 – 2011 (Gt yr⁻¹) | 2012 -2015 (Gt yr⁻¹) | 2016 – 2019 (Gt yr⁻¹) |
|---|---|---|---|---|---|
| Antarctic Peninsula | -19.1±2 | -16.8±1.8 | -24.3±1.9 | -27.2±1.9 | -8.6±2.6 |
| H-Hp | -4.8±1.2 | -2.2±1.1 | -6.2±1.2 | -9.7±1.1 | -1.6±1.3 |
| Hp-I | -3.5±1.3 | 0.4±1.2 | -7.7+1.3 | -9.5±1.2 | 1.7±1.7 |
| I-Ipp | -14.7±1.5 | -16.6±1.3 | -13.5±1.5 | -16.0±1.5 | -12±1.7 |
| Ipp-J | -0.9±1.0 | -0.6±0.8 | -3.1±1.0 | -1.6±1.0 | 1.7±1.1 |

At the individual drainage basin scale (Fig. Figure 4), the same spatio-temporal pattern of process behaviour is evident across both the northern AP (basin I-Ipp), West Palmer Land region (Hp-I) and Abbot sector (H-Hp), with ice dynamics in the northern AP region forming a larger proportion of the overall total imbalance in comparison to the other AP basins (Fig. Figure 4). In contrast, the eastern portion of the southern AP (Ipp-J) has remained in a steady state of dynamic balance for the entirety of the study period, with mass trend fluctuations driven wholly by SMB.



**Figure 3-** SMB and ice dynamic mass trends for the AP region (line plots, with shaded ribbons representing 1σ and 2σ uncertainties). Scatter points represent RCM mass anomalies with respect to a 1979-2002 baseline (Van Wessem et al., 2016; Melchior Van Wessem et al., 2018; Agosta et al., 2019), grey line represents annual ice discharge anomaly with respect to a 1979-2002 baseline value, with the 1σ uncertainty on the discharge represented by the crossbars as stated in (Rignot et al., 2019). See text S1 for methodologies used for dataset intercomparison.






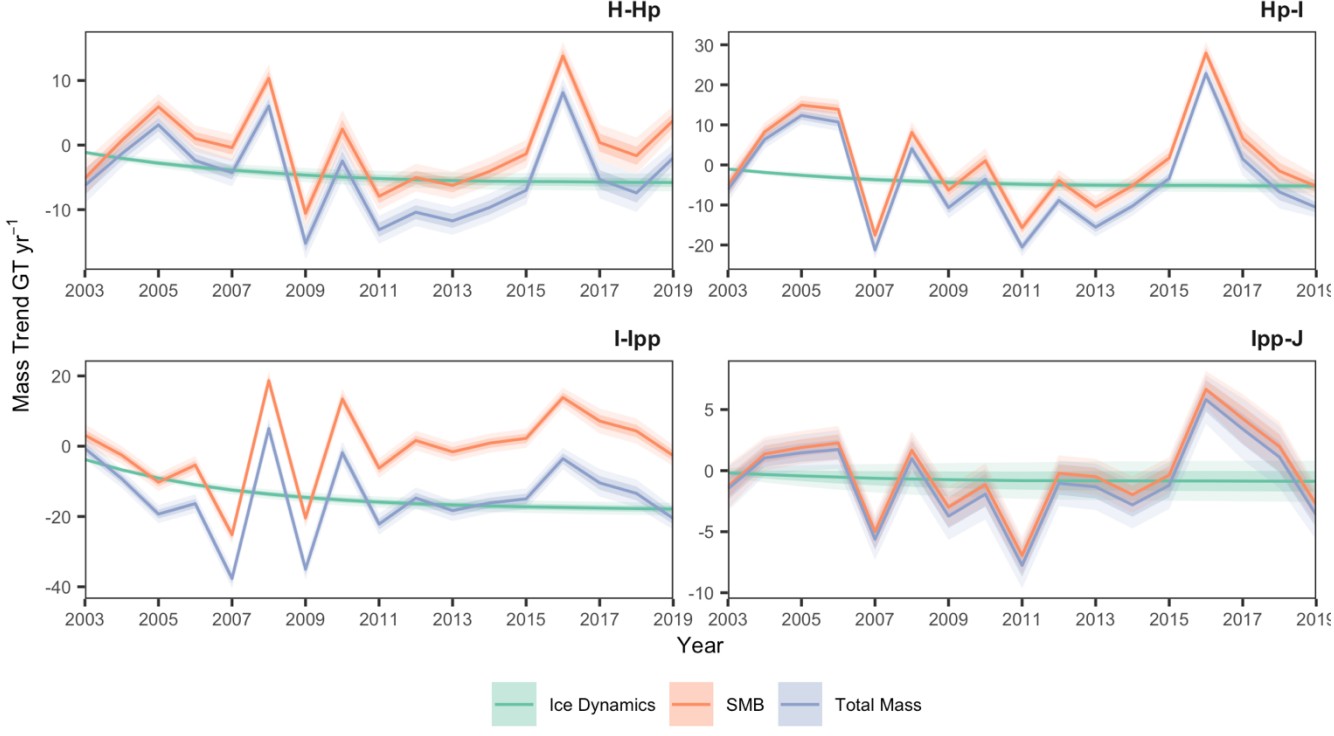

**Figure 4** – BHM mass trends and constituent processes for each individual drainage basin in the study region (Rignot et al., 2011b, 2013), see Fig. 1 for drainage basin locations.. Dark and light ribbons represent the 1σ and 2σ uncertainty of mass trends, respectively.

**4.1 Ice Dynamics**

The temporal pattern of ice dynamic mass changes is similar across both the northern AP (I-Ipp), West Palmer Land (Hp-I) and the Abbot sector (H-Hp) (Fig. Figure 4). Between 2003 and 2010 there is an acceleration in mass loss due to ice dynamics, whereas in the last decade the rate of acceleration either slows or reaches a new state of disequilibrium. As can be seen from Figure 3, for the AP as a whole, comparisons of the BHM ice dynamics mass trend estimated with the ice discharge component of an IOM approach (Rignot et al., 2019) show a similar temporal evolution, albeit with slight negative bias in discharge anomaly with respect to the BHM trends. The congruent nature of the ice dynamics response for basins draining into the Bellingshausen Sea Sector implies an external oceanic forcing mechanism is the likely driver. This is supported by observations showing rapid ice shelf thinning, and therefore reduced buttressing capability to the grounded ice sheet since 1994, with the Venable Ice Shelf in the Bellingshausen Sea Sector showing some of the largest rates of thinning across Antarctica (Paolo et al., 2015). In contrast the eastern side of the AP (basin Ipp-J), draining into the Weddell Sea, shows no change ice dynamics signal according to our BHM estimates. This is consistent with observed deep ocean water temperatures of several degrees above the seawater freezing point in the Bellingshausen Sea Sector, as opposed to the Weddell Sea where they have stayed





near freezing point (Adusumilli et al., 2020). Our ice dynamic trends reflect these contrasting patterns in ocean forcing between the east and west sides of the AP.

In the northern AP, the temporal evolution of ice dynamic imbalance for the period 2001-2010 is consistent with the rapid acceleration and increases in ice discharge of outlet glaciers after the collapse of the Larsen A and B ice shelves in 1998 and
2002, respectively (Scambos et al., 2014), and subsequent deceleration in mass loss (Rott et al., 2018). The Hektoria and Green glaciers draining into the Larsen B embayment show some of the largest thinning rates across the AP, with an estimated -10.1±1.1 m yr$^{-1}$ of the total elevation change signal in 2010 near the grounding line attributed to ice dynamics.

In the West Palmer Land drainage basin (Hp-I), ice dynamic mass loss is concentrated in outlet glaciers draining the English
coast into the George VI ice shelf, the Nitikin Glacier and the Berg and Thompson ice streams, in addition to the dynamic thinning of glaciers draining into the Wordie ice shelf after its collapse in 1990s (Figure 5). Towards the interior of the drainage basin, there is an emergence of a small ice dynamic thickening signal, which is notable due to the slow ice flow in this region. Elevation changes from satellite altimetry have shown the same pattern of thickening over the higher elevation regions of this basin since 2003 (Schröder et al., 2018; Smith et al., 2020). Any long-term temporal trend in ice thickening due to a sustained
increase in precipitation may be partially mis-attributed as an ice dynamic signal by the framework, due to it being a temporally smooth and sustained increase in mass over time. Over the course of the 20th Century there has been sustained large increases in precipitation over the AP, providing an explanation for this misrepresentation (Medley and Thomas, 2019; Wang et al., 2017).

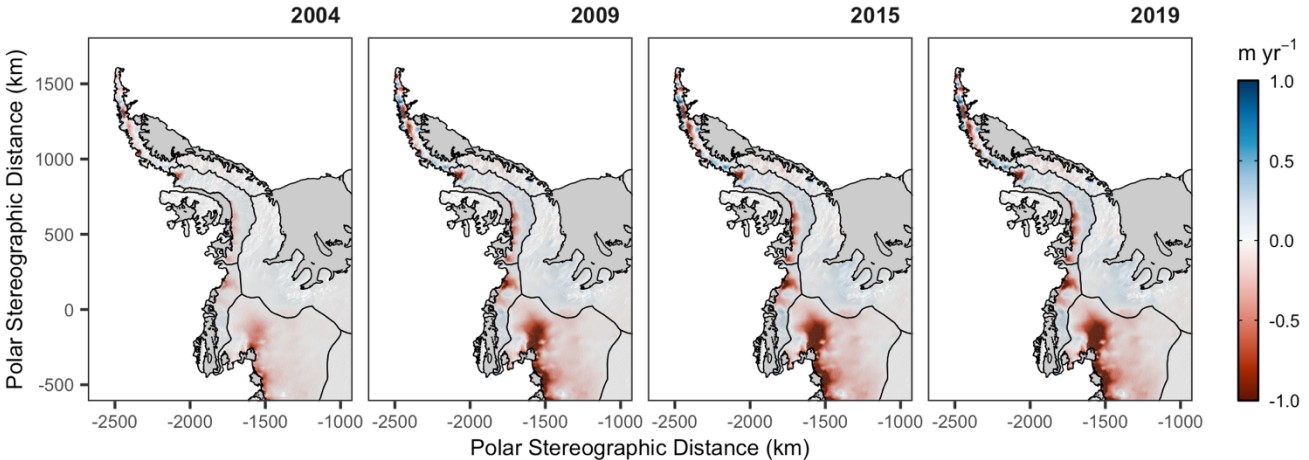

**Figure 5** – Temporal evolution of height changes due to ice dynamics for four years over the course of the 17-year study period. Colour scale is saturated at end points for plot clarity. Basin outlines and grounding line overlain in black (Rignot et al., 2011b, 2013). Dark grey regions represent ice shelves (Depoorter et al., 2013). Data is overlain on the hill shade of the REMA DEM (Howat et al., 2019)



To the west of the AP along the Bellingshausen coastline, the Abbot sector has experienced an increase in ice dynamic mass loss over the period of the study, driven primarily by the Fox, Ferrigno, Wesnet & Williams Ice Streams (Fig. Figure 1a). The ice shelves buttressing these ice streams have seen nearly a one fifth reduction in ice thickness since 1994 (Paolo et al., 2015), which supports the dynamic thinning behaviour seen in the BHM output. This is in direct contrast to the rest of the glaciers in this basin draining into the Abbot Ice Shelf, which have experienced no dynamic induced thinning over the course of the last two decades. This corroborates with other mass budget studies (Chuter et al., 2017) and the absence of grounding line retreat

over several decades (Christie et al., 2016).

**4.2 Surface Processes**

Changes in surface processes is the major driver of inter-annual variability in mass balance, with the total contribution to the mass trend ranging between $-47.7\pm1.8$ Gt yr$^{-1}$ in 2007 to $48.5\pm2.1$ Gt yr$^{-1}$ in 2016 (Figure 3). The largest positive trends in surface processes, in 2016 and 2017, coincide with the extreme El-Niño Southern Oscillation event which resulted in increased

precipitation, most noticeably across the western AP (Bodart and Bingham, 2019). This temporarily restored the total mass trend of the sector to a state of positive balance after sustained negative surface processes mass trends between 2010 and 2015. Over the whole AP sector, our trends in surface mass on the whole follow the inter-annual variations in the output of three RCMs (Fig. 3), with the BHM solution generally showing a weaker signal compared to the RCM output for some years (e.g. 2010 and 2016).


In 2016 and 2017, the BHM allocates the majority of SMB increase in the West Palmer Land region of the southern AP (basin Hp-I), with minimal SMB gains observed on the eastern side (Figure 6). This is in good agreement with the output from the RACMO 2.3p2 5.5 km climate model for 2016, although the RCM output reports much greater and widespread height change. Significant surface process driven height changes are also partially resolved in the northern AP, although without the same

clear separation in signal characteristics along its mountain spine. This is most likely due to the paucity of observations available in this region, despite the increase in observational coverage provided by the CryoSat-2 Swath data. Additionally, the very small width of the region (largely less than 100 km in width) limits the ability of GRACE to observe mass changes in this part of the AP.





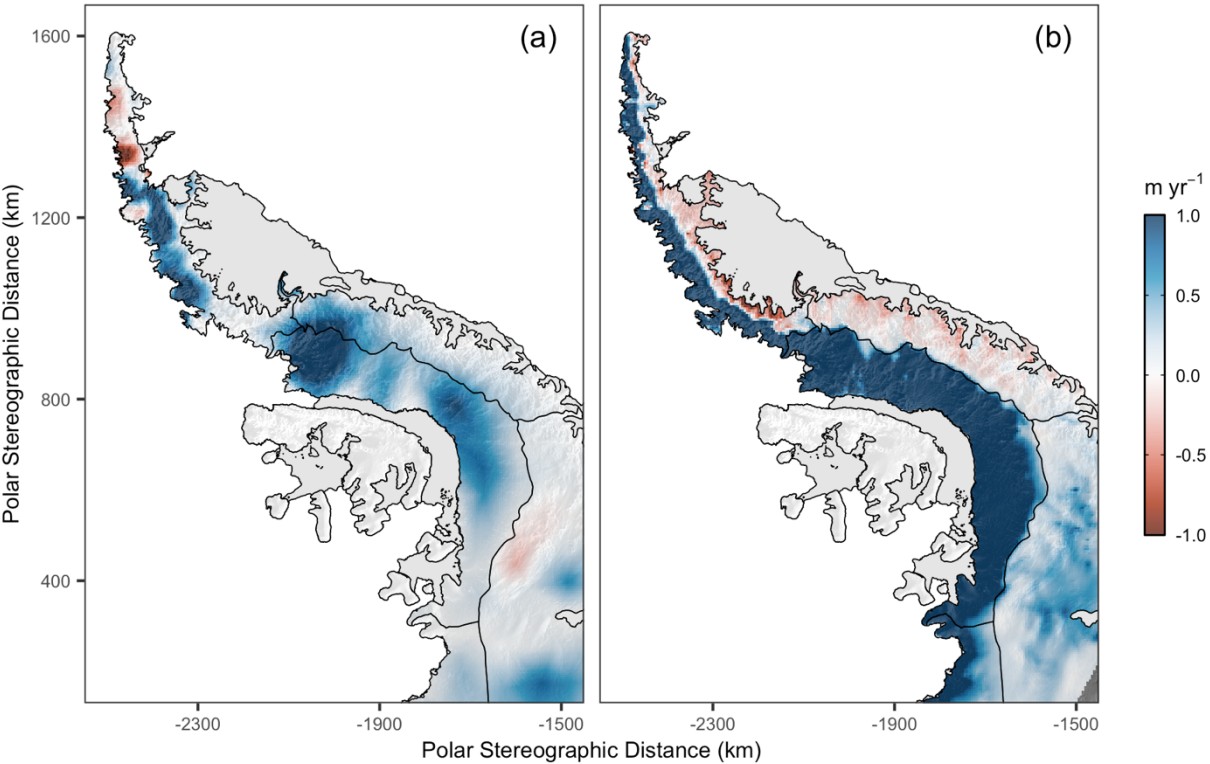

**Figure 6** – Changes in ice sheet elevation due to SMB from the **a)** BHM and **b)** RACMO2.3p2 5.5 km RCM (Van Wessem et al., 2016) for the year 2016. Colour scale is saturated at endpoints for clarity. Basin outlines and grounding line overlain in black (Rignot et al., 2011b, 2013). Dark grey regions represent ice shelves (Depoorter et al., 2013). Data is overlain on the hill shade of the REMA DEM (Howat et al., 2019).

At the individual basin scale, Figure 7 shows that the BHM surface processes output generally follows the temporal variability in the suite of RCMs used for comparison across all basins. This is particularly evident between 2007 and 2011, where the BHM posterior replicates the large positive to negative inter-annual variations in surface processes that are present in all the RCM solutions, albeit with a wide variation in absolute magnitude between both the BHM and each regional RCM. This is expected due to the varying spatial resolutions of the climate models, which govern their capability to resolve small scale orographically-driven processes, in addition to the BHM posterior being driven by observations rather than a physical model of climate processes. The mismatch of magnitude is most prominent in the Hp-I basin, where the RCM output in large anomaly years appears to be consistently either more positive or negative than that of the BHM output.





**Figure 7 -**Comparison of BHM SMB posterior mass trends with RCM output for each year of the study, at the individual basin scale. Black lines represent BHM posterior mean, with dark and light grey boxes representing the 1σ and 2σ uncertainty, respectively, on the latent process. Points represent the output from different RCM's (Van Wessem et al., 2016; Melchior Van Wessem et al., 2018; Agosta et al., 2019). Note that spatial coverage of the RACMO2.3p2 5.5 km model does encompass the full area of basin H-Hp (Abbot sector) and is therefore excluded from analysis in this region. Methodology used for dataset comparison is outlined in text S1.

**5. Discussion**

Our annual mass trend time series for the AP sector shows good general agreement with a variety of time-mean mass balance estimates from different approaches in addition to the IMBIE assessment (Shepherd et al., 2018) (Figure 8). Our time-mean results are in best agreement with altimetry and gravimetry based approaches, though they tend to give more positive mass balance trends than those previously calculated using IOM approaches. This can be partially explained by the fact that our

BHM solution is driven primarily by gravimetry and altimetry observations, so it would be expected that the results would be in better general agreement with these approaches. We also find good overall agreement between our time series and IMBIE for the period 2003-2017 (Shepherd et al., 2018) in terms of the temporal mass trend evolution, showing large variations in mass between 2006 and 2010 and the extreme El Niño driven precipitation event in 2016. On average, our BHM estimate of mass imbalance is slightly more positive than that of the IMBIE time series, which we attribute to the fact that the latter

approach gives equal weighting to estimates from mass budget techniques (which tend to suggest more negative mass balance) in the combined time series. Between 2003 and 2010, the IMBIE study reports negative trends from mass budget approaches that are more than double those calculated using either altimetry and gravimetry, which would propagate into a more negative combined trend (Shepherd et al., 2018). However, the general overall agreement between the two studies demonstrates the value that multiple independent combination approaches can have in improving confidence in mass balance assessments for

this challenging region.





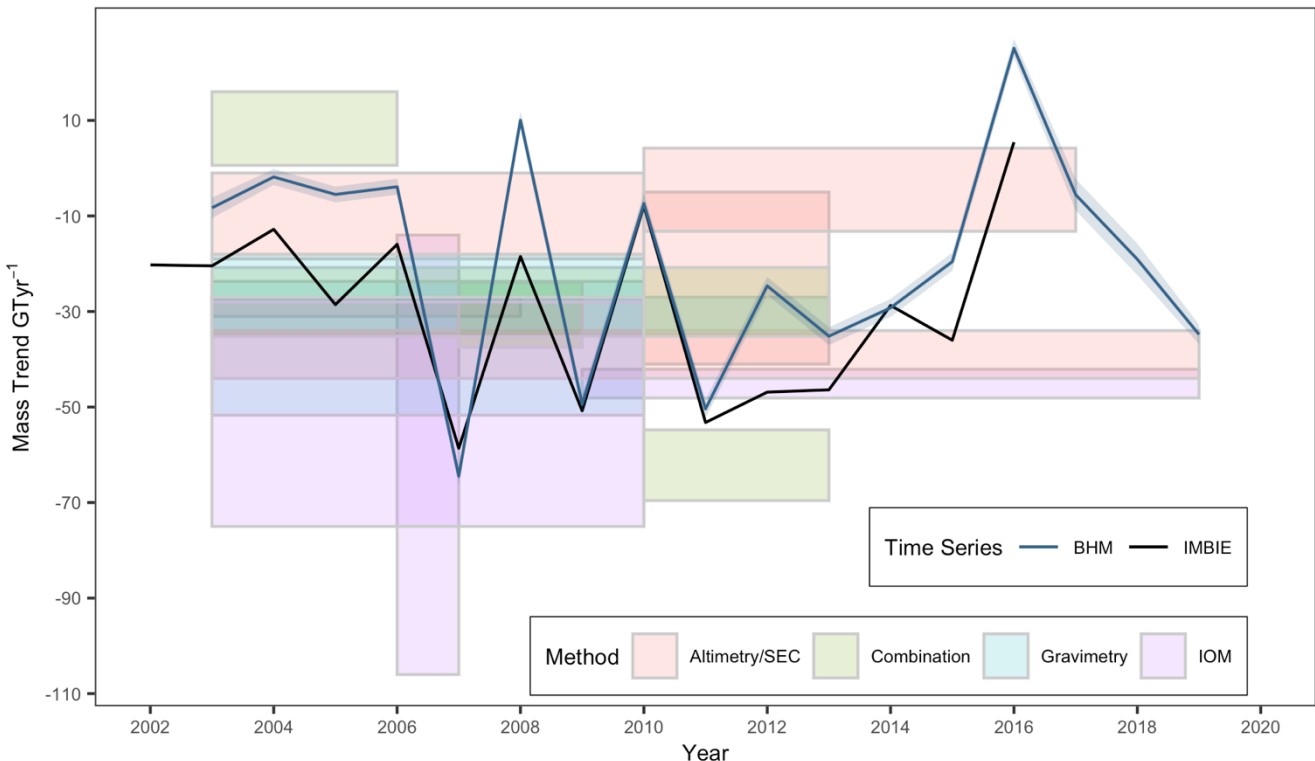

**Figure 8 -** Comparison of total mass trends for the AP from the BHM (blue line, with 1σ uncertainty ribbon) with time averaged epoch studies for the same sector (colour shaded boxes – see Table S1 for list of studies) and the annual mass trends from the IMBIE exercise (Shepherd et al., 2018). Details of the processing for the comparison datasets can be found in Text S1.


When looking at the individual drivers of mass change for the sector as a whole, IOM studies show that the AP has exhibited increases in ice discharge over the last two decades, with an estimated increase in discharge of 14 Gt yr$^{-1}$ between 2003 and 2017 (Rignot et al., 2019) and 7± 6 Gt yr$^{-1}$ between 2008 and 2015 (Gardner et al., 2018). Our ice dynamics solution shows a similar magnitude of ice mass loss over the same period, with an estimated 19.1 Gt yr$^{-1}$ increase between 2003 and 2017 and

an estimated 4.8 Gt yr$^{-1}$ increase between 2008 and 2015. Whilst our rates of change in discharge are generally in agreement with IOM studies, there are differences of over 100 Gt yr$^{-1}$ in the absolute annual ice discharge for contemporaneous time periods over the AP basin (Rignot et al., 2019; Gardner et al., 2018). As a result, calculation of total mass balance using an IOM approach is highly conditional on the choice of baseline. Additionally, it highlights the importance for continuing improvements in the determination of ice thickness and ice velocity at the grounding line. An advantage of the BHM approach

is that it does not require assumptions regarding the balance state of the ice sheet, therefore negating the potential for biases and uncertainties that can occur in the IOM approach.



At the basin scale, reconciling mass balance estimates over the western part of the southern AP (basin Hp-I) for the last two decades has been challenging. For example, two mass balance studies utilising CryoSat-2 radar altimetry and covering the

same period (albeit with differing volume-to-mass conversion philosophies) do not agree within error. The first, using an ice dynamic imbalance mask conversion approach (McMillan et al., 2014) estimated a mass loss of 11±11 Gt yr$^{-1}$, while the second, using the output of regional climate models, estimated a mass loss of 34±12 Gt yr$^{-1}$ (Wouters et al., 2015). For the same 2011-2014 period, our study gives a total mass loss of 13.8±1.2 Gt yr$^{-1}$, consisting of an ice dynamics contribution of -5.0±0.8 Gt yr$^{-1}$ (38%) and a surface processes contribution of -8.8±1.3 Gt yr$^{-1}$ (62%). This time period coincides with the

consistent negative BHM surface processes trend between 2010 and 2015, as shown in Figure 9a, and is in agreement with the negative mass anomalies from RCMs for the same time period. The presence of an ice dynamic thinning signal on the southern sections of the West Palmer Land basin in our solution is consistent with the largest rates of basal melt driven ice shelf thinning over the George VI ice shelf of 7±2 m yr$^{-1}$ for the period 1994-2016 (Adusumilli et al., 2018) (Fig S1a), supporting a joint ice sheet mass loss forcing mechanism for this period of time. Our results suggest that surface processes were the largest

contributor to total mass imbalance between 2011 and 2014, as opposed to ice dynamics. Therefore, it is likely that the use of a dynamic imbalance mask underestimates the ice dynamics contribution (as no areas of this basin was defined a priori to be in an ice dynamics imbalance state), whereas potential biases in the SMB and firn model correction may have underestimated the surface process contribution. This challenge is demonstrated when we compare our time series with other studies over the 17-year period (Figure 9b), for which there is large divergence between time mean mass balance studies since 2010. Whilst

our time series is situated in the middle of the spread of results, different altimetry studies disagree on the sign of mass balance for overlapping periods. This is due in combination to outlined methodological challenges using altimetry observations and to time mean studies not being able to represent the large inter-annual surface process driven variability we see in our mass trend after 2010.

The separation of ocean and ice mass change signals in gravimetry approaches can greatly affect the resultant mass anomaly time series, in particular for the AP and Hp-I basin (Bodart and Bingham, 2019). In terms of basin leakage impacts, other combination approaches with altimetry and gravimetry have shown that the inclusion of the former reduces the corelation in mass change signal between neighbouring basins compared to using gravimetry alone (Sasgen et al., 2019). Considering the Hp-I basin is surrounded by large mass losses from the Abbot and Amundsen Sea sector to the south and west, in addition to

mass losses over the northern AP, the leakage of signals from neighbouring basins are likely to impact gravimetry mass changes in this locality. Additionally the Sasgen et al. (2019) combination study noted the ability for altimetry to observe small spatial mass gain signals due to increased snowfall, which cannot be resolved by gravimetry. As we see in our results for 2016 for this basin (Fig. Figure 6), its susceptibility to large variations in mass from extreme snowfall events may limit agreement between the two observational approaches. Agreement between the two approaches is also possible as the result of a

cancellation of errors.

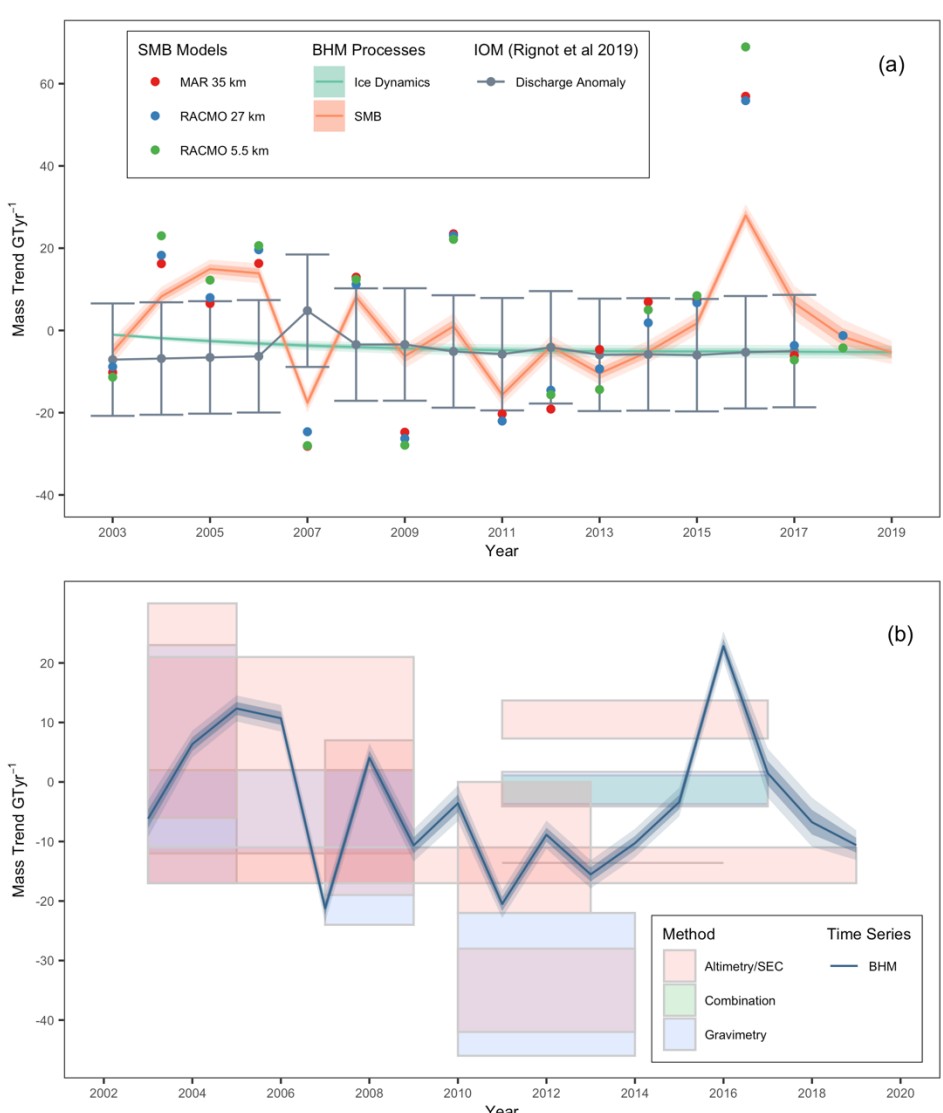

**Figure 9 - a)** Surface processes and ice dynamic mass trends for the Hp-I drainage basin (West Palmer Land and English Coast region) line plots, with shaded ribbons representing 1σ and 2σ uncertainties on the process posterior from the BHM. Scatter points represent RCM mass anomalies with respect to a 1979-2002 baseline (Van Wessem et al., 2016; Melchior Van Wessem et al., 2018; Agosta et al., 2019), grey line represents annual ice discharge anomaly with respect to a 1979-2002 baseline value, with one 1σ uncertainty on the discharge represented by the crossbars (Rignot et al., 2019). **b)** Comparison of total mass trends for basin Hp-I from the BHM (blue line, with 1σ uncertainty ribbon) with time averaged epoch studies for the same sector (colour shaded boxes – see Table S2 for list of studies). Details of the processing for the comparison datasets can be found in Text S1.


Mass budget studies of this basin suggest that the magnitude of ice discharge has been relatively constant, with increases in ice discharge of up to 15 km³ yr⁻¹ over three decades (Hogg et al., 2017; Gardner et al., 2018). This small increase is reflected





in our results, which show an acceleration of ice dynamic mass losses up until 2008, with little change thereafter. This temporal evolution is supported by Gardner et al. (2018) which finds an increase of 2±3 Gt yr$^{-1}$ in ice discharge between 2008 and 2015, in addition to the agreement of our results with ice discharge anomalies from Rignot et al (2019) as seen in Fig. 9a. An analysis of velocity change for 2005-2011to 2011-2017 (Text S2 and Fig. S1b) shows minimal changes across the basin, with highly localised accelerations near the grounding line of some individual outlet glaciers. Whilst this would suggest that mass losses due to dynamic imbalance should increase, the thinning of these outlet glaciers could be offsetting expected increases in discharge. These independent assessments support the magnitude of ice dynamic mass losses we see from the BHM approach in this basin. It should be noted that, unlike in our study, the Wordie Ice Shelf is not included in these IOM basin definitions and our ice dynamic trends may, therefore, show a slight underestimation for the basin. The slight ice dynamic thickening signal we see in our solution in the interior of the basin as result of a long-term snowfall accumulation trend may, in part, be misrepresented as an ice dynamic process and be partially responsible for this discrepancy.

The previous implementation of the BHM framework over this region showed total mass losses between 2010 and 2013 of 39±5 Gt yr$^{-1}$, primarily driven by ice dynamics (Martín-Español et al., 2016). Whilst the basins used in that study included the eastern side of the southern AP, minimal mass fluctuations in this region are not expected to significantly compromise comparisons with our study. We believe the discrepancy between our study and the previous implementation is due in part to the linear temporal ice dynamics constraint applied in previous iterations of the framework. The use of a linear model would not be able to capture accelerations in dynamic thinning in the early 2000s followed by a slowdown in ice discharge imbalance after 2008 that we, and other independent approaches (Rignot et al., 2019; Gardner et al., 2018), have detected in this region. This demonstrates the new ice dynamics process implementation's suitability for multi-decadal studies.

Another benefit of our approach can be seen in 2017 and 2018, when gravimetry observations are not available due to the gap in coverage between the GRACE and GRACE-FO missions. Gravimetry is a key observation in our framework as it is the only available direct observation of mass change and, when combined with altimetry, is critical in aiding source separation of driving processes. However, during this period, the BHM is driven solely from CryoSat-2 altimetry data yet, despite this, the ice dynamics trend over this period retains its smooth variation. Surface processes mass trends also remain in good general agreement with temporal variations in RCM output (Figure 3 & Figure 7), albeit with slightly larger uncertainty bounds. The ability to successfully bridge this gap is a key practical demonstration for the first time of one of the major benefits of our BHM approach. It provides confidence that our combination approach can handle future potential unforeseen gaps or other changes in either the altimetry or gravimetry observation record, maintaining the long-term continuous time series required to ascertain the driving forces of ice sheet mass balance.



## 6. Limitations and Future Outlook

Despite the considerable progress made in this study, we can identify several limitations and potential future improvements with our implementation of a BHM framework. First, the validity of the BHM approach in mass balance studies largely hinges on our assumptions about the underlying latent processes and priors being representative of the geophysical processes being solved. In previous implementations of the BHM framework, assuming a linear or quadratic temporal variation in ice dynamics was generally adequate due to the short length of the time series (< 10 years). The change in implementation to an AR(1)

process with drift for ice dynamics in this study addresses a potential issue with extending the time series over multiple decades. However, the high coefficient used on the AR(1) process means the underlying assumption remains that the process is temporally smooth, similar to what we observe in the output of ice sheet models. Therefore, any sudden changes in ice dynamics (sub-annual to annual), such as resulting from sudden ice shelf collapse, may not be represented immediately in this latent process and may take several years to be appropriately attributed. This modelling assumption is apparent in the temporal

smoothness properties of our ice dynamics mass trends in comparison with variability seen in the ice discharge component of mass budget approaches (Figure 3 & Figure 9a) (Rignot et al., 2019). Future priors can address this by taking advantage of recent ice sheet wide repeat velocity measurements (Fahnestock et al., 2015), and allowing for the implementation of a spatio-temporal prior linking ice dynamics to velocity.

Second, despite the increases in observation coverage and mesh density, challenges remain separating latent processes in the northernmost sectors of the AP. This is in part due to the basin's latitudinal position reducing the spatial density of coverage available from satellite altimetry, as well as its small spatial scale relative to the native resolution of GRACE. The incorporation of dDEM data into the framework, although spatially limited in its coverage, alleviates this issue somewhat and demonstrates the potential for improving mass change estimates in the future, as exemplified over the Hektoria and Green glaciers. The

recent launch of ICESat-2, in addition to more widespread repeat dDEM data from the REMA project, should greatly enhance our ability to monitor the mass balance of this key region into the next decade. The spatial length scale of the processes we solve for are limited by the resolution of the prior information used in the model. Therefore, in regions as heterogenous as the northern AP, improved resolution climate models and statistical downscaling approaches (Noël et al., 2016) will aid small spatial scale process separation.


   Third, many of the observations used in the framework are from multi-year trends, necessitated by the satellite repeat revisit time and to ensure robust trend estimation. However, this can, in effect, apply a low-pass filter to the observations, which may reduce inter-annual changes in SMB seen by the framework. Whilst the inter-annual patterns in surface processes mass trend variation are the similar between the model outputs and geophysical models (Figure 7), the BHM posterior field appears to

have a reduced amplitude compared to that of the geophysical models in some basins (e.g. Basin Hp-I), which would be indicative of the observations being used. This caveat should therefore be considered in the interpretation of these fields. Whilst



higher temporal resolution sensors will allow multi-annual trend estimation, it still presents a challenge when combining diverse observations with differing spatio-temporal observation characteristics (Shepherd et al., 2018).

Finally, recent advances in our understanding of the much more rapid than previously assumed (decadal length scales) viscous response of the mantle to changes in ice mass loading for certain sectors of Antarctica (Nield et al., 2014; Barletta et al., 2018) presents a future opportunity for the framework. Incorporation of a temporally evolving latent process for GIA, coupled with GPS observations over the whole study period, could provide the opportunity for a temporally varying data-driven inverse GIA field. Although GIA is not a major contributor to mass changes in the AP, and therefore not a significant source of

uncertainty in this study, such a solution would provide insights into the solid Earth response and its potential impact both on ice sheet stability (Larour et al., 2019) and variations in local sea level change around the globe.

**7. Conclusion**

In this study we estimate the mass evolution of the AP, in addition to its driving constituent processes, at an annual resolution for the period 2003-2019 using a BHM framework. Our results indicate that the sector experienced an overall mean mass loss of 19.1±2 Gt yr$^{-1}$ in this time, driven by an acceleration in ice dynamic imbalance over the first decade of this century and continuing at a lower rate thereafter until present. In contrast, no trends in surface processes are evident over the period of the study, although at annual time scales, they still govern the mass balance state of the region. Our solution provides good

agreement with a range of RCM SMB estimates at both the sector and individual basin scale, reflecting the main inter-annual variability of the surface processes mass trends. Additionally, our ice dynamic mass losses support the general trends in ice discharge seen in independent mass budget studies. We find in the West Palmer Land region (Hp-I) our solution indicates that surface processes are a more likely driver of the large mass losses seen between 2011 and 2014, highlighting the challenges and improvements in the necessary assumptions that need to be made in the volume-to-mass conversion when using

conventional altimetry approaches, particularly when multiple driving processes are occurring in spatially coincident locations. Additionally, addressing the GRACE/GRACE-FO gap highlights the benefit of a combination approach in ensuring the continuation of high-resolution mass change time series over the course of multiple decades, which is crucial for attributing the role of differing external forcing in driving ice sheet mass change. The enhanced BHM framework presented in this study, in combination with emerging state-of-the-art high resolution Earth observations such as WorldView and ICESat-2, lays the

foundation for the extension of the approach over the next decade.
**Code Availability**

Finite Element Meshes used in this study were constructed using the R-INLA software package (https://www.r-inla.org). Bayesian Inference was undertaken using the freely available MVST(V2.0) R package developed by AZM & SJC
(https://github.com/andrewzm/MVST).

**Data Availability**

The Bayesian Hierarchical Model annual mass trends, including constituent processes, are provided in supplementary Dataset S1. The spatial fields used for the visualisations in this manuscript are provided in Dataset S2. Cryosat-2 Swath data used in
this study were processed by CryoTEMPO-EOLIS (https://www.cryotempo-eolis.org) and distributed by ESA (https://science-pds.cryosat.esa.int/). Ice shelf net mass balance data from Adusumilli et al (2018) is available from https://sealevel.nasa.gov/data/dataset/?identifier=SLCP_AP_iceshelf_mass_balance_1. The MEaSUREs annual velocity maps used are available at http://dx.doi.org/10.5067/9T4EPQXTJYW9.

**Author contribution**

SJC conceptualised the study, undertook the analyses and wrote the manuscript. AZM developed the original MVST R software package and assisted SJC in implementation of model improvements. GD provided significant discussions on altimetry processing and swath data analysis. JR advised on the statistical inference framework and the methodology. JLB assisted in methodological discussions and the analysis of results. All authors discussed the results and provided comments on
the manuscript.

**Acknowledgement**

This work was supported by European Research Council (ERC) under the European Union's Horizon 2020 research and innovation programme under grant agreement No 694188 (GlobalMass). Andrew Zammit-Mangion's research was supported by an ARC Discovery Early Career Research Award (DECRA) DE180100203. We would like to thank Ted Scambos for
providing the dDEM and ICESat-1 data used in this study over the northern AP. We also acknowledge Melchior Van Wessem for providing the RACMO2.3p2 SMB and surface density products used in this study.

**Competing interests**

The authors declare that they have no conflict of interest.



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
