# Peer review of "Mass evolution of the Antarctic Peninsula over the last two decades from a joint Bayesian inversion"

_The Cryosphere, 2021_

## Referee Comment (RC2)

**1  Overview**

Chuter et al. (2021) use a joint Bayesian inversion technique and remote sensing datasets to separate the variations of ice sheet processes on an annual scale for the Antarctic Peninsula. This is an advancement of prior work using similar methods for analyzing the surface and mass change of the Antarctic ice sheet as a whole. The authors use higher resolution and more localized datasets to solve for this complex region of ice sheet mass change. The work presented by the authors falls within the scope of *The Cryosphere* and could make an interesting contribution to mass change estimates for the Antarctic Peninsula. Overall, this is a good study with established techniques to analyze a difficult region of Antarctica.

**2  Broad comments**

- The authors note a paucity of altimetry available for the Antarctic peninsula. Could surface elevation measurements data from Operation IceBridge help here?

- What impact does using interpolated gridded products for altimetry have here versus using an along-track product?

- Should use "ICESat" for the original mission rather than "ICESat-1"

- The ENVISAT section (2.2.1) should be after the ICESat section since ENVISAT is "providing complementary coverage to ICESat"

- The MEaSUREs ice velocity and RACMO2.3p2 5.5km descriptions should be in their own subsections.

- Could note that all datasets were converted to a common reference frame and projection versus repeating.

- En dashes should be used for date ranges rather than hyphens

- Colors on the plots could be changed to be able to be discernible when printed

**3  Line-by-line comments**

**Page 1, Lines 11–12:** This makes it seem like it was continual and accelerating losses. However, as noted later in the abstract there were a couple of years of mass gains late in the study period.

**Page 1, Lines 21–22:** Are there any times when surface mass balance and ice dynamics are not happening simultaneously?

**Page 1, Line 28:** Partial or near-full collapse of Larsen-B

**Page 1, Line 32:** I would just say "Continuous long-term monitoring of the AP is important as..."

**Page 2, Line 36:** Remove "Therefore"

**Page 2, Lines 37–38:** I would break this into pieces to be similar to the following: "The Antarctic Peninsula is difficult to monitor using conventional mass balance approaches. The high relief of the peninsula make it a particularly poor region to monitor with satellite altimetry, Interferometric SAR-derived (InSAR) velocities and ice thickness data. In addition, the sharp topography and large climatic gradient across the AP provides significant challenges for predictions from regional climate models. Finally, the more northerly latitude increases the across-track spacing between orbits for satellite altimetry, which decreases the available measurements in the region."

**Page 5, Line 93:** Should note here that these are "a $0.5° \times 0.5°$ global grid interpolated from equal area $3°$ diameter spherical caps."

**Page 5, Line 104:** the mascon delineations are available from JPL

**Page 5, Lines 108–110:** There are larger uncertainties in the GRACE products starting October 2016 due to efforts to extend the mission lifetime and preserve battery life. GRACE-FO also has larger uncertainties due to accelerometer issues on GF2.

**Page 6, Line 144:** averaged and interpolated using a hypsometric approach

**Page 7, Line 171:** is 5km grid spacing be too large for a planar assumption in the AP?

**Page 7, Line 173:** would the 15 m/yr cutoff eliminate some valid points near the grounding line? The near ground-line thinning on Page 15 was 10.1 m/yr.

**Pages 7–8, Lines 179–181:** This is an important piece of the paper and should probably be in the abstract or conclusion.

**Page 8, Line 194:** should be "photogrammetric" acquisitions correct?

**Page 8, Lines 208–211:** A difficulty here would be the sensitivity of GRACE/GRACE-FO to near-grounding line change. If these values are masked in the DEM fields but of a large enough magnitude to be detectable by GRACE/GRACE-FO, then there has to be some sort of bias being introduced. Would be good to have some clarifying text in the discussion.

**Page 9, Line 216:** GIA is "quasi-invariant" over short time scales (years to decades). It would have been interesting to see the results if GIA wasn't fixed as the uncertainty is a limiting factor in GRACE/GRACE-FO studies (∼4–5 Gt/yr difference between Ivins et al. (2013) and Whitehouse et al. (2012) in the AP)

**Page 11, Line 265:** Is this 99.6% of the AP or ice sheet wide?

**Page 11, Lines 274–276:** For overlapping grid cells in the 27km and 5.5km models, are the average spectral properties the same?

**Page 11, Line 277:** Use something besides "per se"?

**Page 11, Lines 287–289:** The near collapse of Larsen-B resulted in rapid sub-annual responses in ice velocity (Rignot et al., 2004). Would this still be over-smoothing potential discharge contributions?

**Page 14, Line 324:** Figure 4 instead of "Fig. Figure 4"

**Page 16, Line 367:** "Changes in surface processes are the major drivers"

**Page 16, Line 380:** Could other altimetry datasets (OIB/IS2) help with the "paucity" of data?

**Page 16, Line 382:** If it is a large enough signal in magnitude, GRACE/GRACE-FO should be able to observe it. However, it would be a smoothed representation of that mass change and difficult to localize due to the non-uniqueness of the solutions.

**Page 19, Lines 413–415:** What about comparing to the individual components of IMBIE?

**Page 21, Lines 467:** "cannot be resolved with gravimetry alone."

**Page 23, Lines 486:** Is there a space missing between "2011" and "to"?

**Page 23, Lines 498–499:** Would the deficiencies in calculating ice discharge in the Martín Español et al. (2016) solutions affect the GIA solution used here?

**References**

S. J. Chuter, A. Zammit-Mangion, J. Rougier, G. Dawson, and J. L. Bamber. Mass evolution of the Antarctic Peninsula over the last two decades from a joint Bayesian inversion. *The Cryosphere Discussions*, 2021:1–32, 2021. doi: `10.5194/tc-2021-178`.

E. R. Ivins, T. S. James, J. Wahr, E. J. O Schrama, F. W. Landerer, and K. M. Simon. Antarctic contribution to sea level rise observed by GRACE with improved GIA correction. *Journal of Geophysical Research: Solid Earth*, 118(6):3126–3141, June 2013. ISSN 2169-9356. doi: `10.1002/jgrb.50208`.

A. Martín Español, A. Zammit Mangion, P. J. Clarke, T. Flament, V. Helm, M. A. King, S. B. Luthcke, E. Petrie, F. Rémy, N. Schön, B. Wouters, and J. L. Bamber. Spatial and temporal Antarctic Ice Sheet mass trends, glacio-isostatic adjustment, and surface processes from a joint inversion of satellite altimeter, gravity, and GPS data. *Journal of Geophysical Research: Earth Surface*, 121(2):182–200, Feb. 2016. ISSN 2169-9011. doi: `10.1002/2015JF003550`. 2015JF003550.

E. J. Rignot, G. Casassa, S. P. Gogineni, W. B. Krabill, A. Rivera, and R. H. Thomas. Accelerated ice discharge from the Antarctic Peninsula following the collapse of Larsen B ice shelf. *Geophysical Research Letters*, 31 (L18401):1–4, 2004. doi: `10.1029/2004GL020697`.

P. L. Whitehouse, M. J. Bentley, G. A. Milne, M. A. King, and I. D. Thomas. A new glacial isostatic adjustment model for Antarctica: calibrated and tested using observations of relative sea-level change and present-day uplift rates. *Geophysical Journal International*, 190(3):1464–1482, 2012. doi: `10.1111/j.1365-246X.2012.05557.x`.

---

## Author Comment (AC1)

Dear Dr. Stef Lhermitte,

We would like to thank the two reviewers for their insightful and helpful comments, which have helped us improve and clarify the manuscript where required. In our response, original comments from the reviewer are shown in blue, with our response shown below in black. Amendments to the text are shown in grey italics with associated line numbers referring to the original manuscript submitted for peer review. We hope you find these changes to the manuscript satisfactory for publication in The Cryosphere.

Kind regards,

Stephen Chuter (on behalf of all co-authors)

**Reviewer #1**

This manuscript presents the results of a Bayesian hierarchical model (BHM) applied to the mass balance of the Antarctic peninsula. The technique combines gravimetry data and a variety of altimetry data with statistical constraints on the spatial patterns of change based on velocity and RCM data to derive estimates of glacier mass changes and of the processes leading to those changes. The authors have performed this kind of analysis at a variety of scales on a few different regions, and have their science and presentation, as it were, down to a science. The manuscript shows a high degree of polish, and I had very few editorial or scientific concerns with it. The results are largely in line with other studies of the mass balance of the region, and the authors have reasonable explanations for any differences between their estimates and those in the literature.

We thank the reviewer for their positive comments regarding the manuscript's methodological approach and scientific outcomes.

My only misgiving about how the BHM results were in the estimates of the SMB shown in figure 6. It is somewhat apparent from this figure that the available data are not adequate to resolve the spatial pattern of the surface processes correctly for at least the year 2016, let alone their magnitude. The authors acknowledge that this is likely due to the small width of the region and the lack of data, but I would have expected this to result in larger error estimates in figure 7. Is it possible that the prior model for SMB variability is too smooth in this part of the AP? The RACMO gradient appears to be quite sharp, and it was not clear to me whether prior would allow a steep gradient in this specific part of the Peninsula.

We acknowledge that there can be differences between the RCM's and the model posterior (as shown in Fig 6), which are due to a variety of factors including: observational coverage, temporal baseline of the observations and differing methodological approaches in determining the process magnitude between an RCM and the BHM– one is data driven and the other is a geophysical model simulation. The altimetry trends derived from a multi-annual baseline by the BHM also serve to low-passlow pass filter the SMB signal (as acknowledged on Lines 540-543 of the submitted manuscript), so a certain level of smoothness, when compared to the RCM output, is expected.

The aim of this figure was to show that, even for 2016 where there is an exceptionally large SMB signal in the RCM (in context of the time period of the study), the main patterns of positive SMB trends are resolved by the BHM over the West Palmer Land and Wordie Ice Shelf region of the Hp-I basin and is broadly represented over the Northern Antarctic Peninsula (although we acknowledge that the distinct separation of signals East to West is not as well reflected in our model over this challenging region). The prior spatial length scale for the SMB variability is derived from fitting a

Gaussian spatio-temporal separable model to the region in blocks (Zammit-Mangion et al., 2015). The typical spatial length scale is ~140km, small enough to capture variations across the East to West topographic gradient of the Southern Antarctic Peninsula (~200km width), whilst maintaining enough smoothness to represent the homogenous SMB induced elevation increases seen over West Palmer Land in the RCM's if present in the data. Additionally for the AP region, the BHM has previously been shown to be insensitive to changes in SMB length scale (Martín-Español, Zammit-Mangion, et al., 2016; Schoen et al., 2015). The smoothness can also be partly attributed to the observation resolution, for which in the case of GRACE is coarse in this region of the ice sheet. A future model development to address such smoothness, could be the introduction of a barrier model into the SMB process to enforce spatial independence in the solution between the East and West sides of the Antarctic Peninsula. This would spatially constrain the process more effectively, although the overall process smoothness would still be dependent on the factors outlined above. We have added this future research avenue to the text:

[Line 539] *"An additional future model development could be the introduction of a barrier model to enforce spatial independence in regions which, whilst in close spatial proximity, are largely spatio-temporally independent. This would be useful to capture, for example, the differing SMB characteristics between the East and West sides of the Antarctic Peninsula Mountain range."*

With regards to uncertainties, the fact that the aggregates (Figure 7) are more certain than the marginals (i.e., the pointwise spatial uncertainty) is an indication that our posterior distributions are highly spatially (anti-)correlated. That is, the uncertainty at a given spatial point is quite high, but when aggregated over a large region can be quite small. This is a result from the relationship var(A + B) = var(A) + var(B) - 2cov(A,B) for two random variables, A,B: If cov(A,B) is sufficiently negative, it is possible (and quite likely when one assumes smooth spatial processes and when observations have a large support) that var(A + B) < min(var(A), var(B)). To show that our marginal uncertainties are larger, we include in this response a figure of the posterior prediction standard deviation over the AP, which has also been added as a supplementary figure (Figure S1). The spatial posterior standard deviation fields are also included in the supplementary dataset to this manuscript (Dataset S2).

[Figure]

[Figure]

*Figure S1 -* **Left)** *Posterior standard deviation of the SMB spatial field for the year 2010.* **Right)** *Posterior Standard deviation of the ice dynamics process posterior for the year 2010. Basin outlines* (E Rignot et al., 2011, 2013)*, grounding line and ice shelf outlines are shown in black (Depoorter et al., 2013)*.

This new supplementary figure has been referred to in the main manuscript in the following manner:

[Line 397] *"The basin aggregated SMB posterior magnitudes are more certain than the marginal (spatial pointwise) posterior standard deviations, due in part to the relative smoothness of the process and the large spatial resolution of some of the observations. An example of the posterior standard deviations associated with our predictions for SMB and ice dynamics for the year 2010 are shown in Fig. S1 and in Dataset S2"*

Editorial comment (sorry, just the one)

Line 507: add a comma before 'yet,' and consider more drastic measures on this sentence

Agreed the sentence was poorly worded. Have amended to the following:

[Line 507] *"However, despite the BHM being solely driven by CryoSat-2 altimetry data during this period, the BHM ice dynamics posterior trend over this period retains its smooth variation"*

**Reviewer #2**

1 Overview

Chuter et al. (2021) use a joint Bayesian inversion technique and remote sensing datasets to separate the variations of ice sheet processes on an annual scale for the Antarctic Peninsula. This is an advancement of prior work using similar methods for analyzing the surface and mass change of the Antarctic ice sheet as a whole. The authors use higher resolution and more localized datasets to solve for this complex region of ice sheet mass change. The work presented by the authors falls within the scope of The Cryosphere and could make an interesting contribution to mass change estimates for the Antarctic Peninsula. Overall, this is a good study with established techniques to analyze a difficult region of Antarctica.

We thank the reviewer for their positive comments on the novelty and the contribution of our manuscript to this field of Antarctic Peninsula mass balance.

Broad comments

The authors note a paucity of altimetry available for the Antarctic peninsula. Could surface elevation measurements data from Operation IceBridge help here?

OIB data was looked at during the development of this study, but coverage was very limited pre-2010 and substantial coverage (beyond a handful of flight lines) was mainly present over the 2014-2016 period over the Southern AP. Given the much more comprehensive coverage of the CryoSat-2 swath data over the same period for that region, it was deemed to have limited extra value in this instance. However, it is something that can be possibly incorporated into future model runs over the ice sheet.

What impact does use interpolated gridded products for altimetry have here versus using an along-track product?

The impact of using gridded products versus along track for altimetry data is negligible, mainly due to the finite element mesh approach our model employs. Observations that are of higher resolution than the minimum mesh element (~5 km in this case for ice dynamics) would be implicitly smoothed over by the mesh coarseness at run time if passed to the model framework as an along track product. Additionally, gridding makes the data set smaller in size, and fitting the BHM

computationally efficient (this is particularly useful for large datasets such as the CryoSat-2 swath data); gridding/aggregation also helps filter outliers that may be present in the raw along-track elevation change product.

Should use "ICESat" for the original mission rather than "ICESat-1"

Agreed, reworded throughout the text.

The ENVISAT section (2.2.1) should be after the ICESat section since ENVISAT is "providing complementary coverage to ICESat"

Agreed, we have re-arranged and renumbered the sections accordingly.

The MEaSUREs ice velocity and RACMO2.3p2 5.5km descriptions should be in their own subsections.

These paragraphs of the text have now been given their own subsection. Additionally, they have been moved to the end of Section 3 after Line 293, to enhance readability (as the changes in ice dynamic processes are associated with the model outline section at the start, given the new structure)

Could note that all datasets were converted to a common reference frame and projection versus repeating.

Agreed, as it avoids unnecessary repetition. We have now removed projection information for each observation and have instead added the following on Line 72:

[Line 72] *"All datasets were then converted to a common Antarctic Polar Stereographic Projection (EPSG:3031)"*

 Em dashes should be used for date ranges rather than hyphens

Thank you, all hyphens have now been replaced throughout the manuscript with En dashes to denote date ranges as per the The Cryosphere manuscript guidelines.

Colors on the plots could be changed to be able to be discernible when printed

We agree with the reviewer that there were some plots that, whilst discernible on screen, were not as clear when printed. After printing and going through the manuscript, we identified these as figures 3,4,8,9. We have adjusted the figures to using a darker colour palette (Dark2 from colorbrewer.org), in addition to changing the discharge anomaly line plots to black. We have also reduced the alpha transparencies used to represent uncertainty values in time series plots and for box plot comparisons. The new figures along with captions can be seen below:

[Figure]

*Figure 3- SMB and ice dynamic mass trends for the AP region (line plots, with shaded ribbons representing $1\sigma$ and $2\sigma$ uncertainties). Scatter points represent RCM mass anomalies with respect to a 1979–2002 baseline* (Agosta et al., 2019; Melchior Van Wessem et al., 2018; Van Wessem et al., 2016)*, black line represents annual ice discharge anomaly with respect to a 1979–2002 baseline value, with the $1\sigma$ uncertainty on the discharge represented by the crossbars as stated in* (Eric Rignot et al., 2019)*. See text S1 for methodologies used for dataset intercomparison.*

[Figure]

*Figure 4 – BHM mass trends and constituent processes for each individual drainage basin in the study region* (E Rignot et al., 2011, 2013)*, see Fig. 1 for drainage basin locations.. Dark and light ribbons represent the 1σ and 2σ uncertainty of mass trends, respectively.*

[Figure]

*Figure 8 - Comparison of total mass trends for the AP from the BHM (blue line, with 1σ uncertainty ribbon) with time averaged epoch studies for the same sector (colour shaded boxes – see Table S1 for list of studies) and the annual mass trends from the IMBIE exercise (Shepherd et al., 2018). Details of the processing for the comparison datasets can be found in Text S1.*

[Figure]

*Figure 9 - a) Surface processes and ice dynamic mass trends for the Hp-I drainage basin (West Palmer Land and English Coast region) line plots, with shaded ribbons representing 1σ and 2σ uncertainties on the process posterior from the BHM. Scatter points represent RCM mass anomalies with respect to a 1979–2002 baseline* (Agosta et al., 2019; Melchior Van Wessem et al., 2018; Van Wessem et al., 2016)*, black line represents annual ice discharge anomaly with respect to a 1979–2002 baseline value, with one 1σ uncertainty on the discharge represented by the crossbars* (Eric Rignot et al., 2019). *b) Comparison of total mass trends for basin Hp-I from the BHM (blue line, with 1σ uncertainty ribbon) with time averaged epoch studies for the same sector (colour shaded boxes – see Table S2 for list of studies). Details of the processing for the comparison datasets can be found in Text S1.*

Page 1, Lines 11–12: This makes it seem like it was continual and accelerating losses. However, as noted later in the abstract there were a couple of years of mass gains late in the study period.

This sentence was not intended to convey this meaning, instead it simply meant to state that on average over the last two decades the AP region has seen its mass losses increase (as noted in the IMBIE intercomparison exercises (Shepherd et al., 2018)). We have now reworded this to improve clarity:

[Line 11-12] *"The Antarctic Peninsula has become an increasingly important component of the Antarctic Ice Sheet budget over the last two decades, with mass losses generally increasing"*

Page 1, Lines 21–22: Are there any times when surface mass balance and ice dynamics are not happening simultaneously?

Yes, they are always occurring simultaneously. We have now removed the phrase "although both processes are acting simultaneously" in the sentence, which suggested that this simultaneous behaviour was only the case for the early part of the 2010s:

[Line 21-22] *"In the West Palmer Land and the English Coast regions, surface processes are a greater contributor to mass loss than ice dynamics in the early part of the 2010s"*

Page 1, Line 28: Partial or near-full collapse of Larsen-B

Done. The sentence now reads as:

[Line 28] *"The collapse of the Larsen A and near-full collapse of Larsen B ice shelves in 1995 and 2002"*

Page 1, Line 32: I would just say "Continuous long-term monitoring of the AP is important as. . . "

Agreed, reworded as suggested

Page 2, Line 36: Remove "Therefore"

Done.

Page 2, Lines 37–38: I would break this into pieces to be similar to the following: "The Antarctic Peninsula is difficult to monitor using conventional mass balance approaches. The high relief of the peninsula make it a particularly poor region to monitor with satellite altimetry, Interferometric SAR-derived (InSAR) velocities and ice thickness data. In addition, the sharp topography and large climatic gradient across the AP provides significant challenges for predictions from regional climate models. Finally, the more northerly latitude increases the across-track spacing between orbits for satellite altimetry, which decreases the available measurements in the region."

We agree with the reviewer's suggestion and have replaced the text on lines 37-41 with the following:

[Line 37-41] *"The AP is difficult to monitor using conventional mass balance approaches. The high relief of the peninsula makes it a particularly poor region to monitor with satellite altimetry, Interferometric SAR-derived (InSAR) velocities and ice thickness data. In addition, the sharp topography and large climatic gradient across the AP provides significant challenges for regional climate models. Finally, the more northerly latitude increases the across-track spacing between orbits for satellite altimetry, which decreases the available measurements in the region."*

Page 5, Line 93: Should note here that these are "a 0.5∘×0.5∘ global grid interpolated from equal area 3∘ diameter spherical caps."

We have incorporated this amendment into the text into L93 as follows:

[Line 93] *"Posted on a 0.5° x 0.5° degree global grid interpolated from equal area 3∘ diameter spherical caps."*

Page 5, Line 104: the mascon delineations are available from JPL

We have added this clarification to the end of this sentence L104:

*"(mascons delineations are available from JPL)."*

Page 5, Lines 108–110: There are larger uncertainties in the GRACE products starting October 2016 due to efforts to extend the mission lifetime and preserve battery life. GRACE-FO also has larger uncertainties due to accelerometer issues on GF2.

We agree with the reviewer regarding the uncertainties in the GRACE product and have amended lines 108-110 to reflect this:

*"Due to the temporal gap between degradation of the GRACE satellite in October 2016 and the launch of GRACE-FO in May 2018, trends for the calendar years 2017 and 2018 were not estimated. Additionally, GRACE-FO observations are expected to be more uncertain due to issues with the instrument's accelerometer."*

Page 6, Line 144: averaged and interpolated using a hypsometric approach

This would not be accurate– the ICESat elevation trends are not averaged in accordance with distinct elevation bands, purely in an along track manner. The quasi-regular grid averaging accounts for the potential uneven distribution of tracks within a grid cell and therefore the term interpolation would not be strictly appropriate in this context. As a result, we have maintained the original text.

Page 7, Line 171: is 5km grid spacing be too large for a planar assumption in the AP?

Our approach to determining CryoSat-2 $\frac{\Delta h}{\Delta t}$ estimates is well established in the altimetry community, with a planar 5 km grid scale being used over the Antarctic Peninsula in previous literature (McMillan et al., 2014; Shepherd et al., 2019). Additionally, it was an appropriate size given the resolution of finite element mesh being used in this study. The plane fit approach accounts for variations in topography within the grid cell through the determination of coefficients $a_1, a_{2,} a_3$ in Equation 2 of the manuscript. What we have now done is added the references mentioned in this response to L171 after the discussion of grid cell sizing, to acknowledge the choice of this parameter in relation to the literature.

Page 7, Line 173: would the 15 m/yr cutoff eliminate some valid points near the grounding line? The near ground-line thinning on Page 15 was 10.1 m/yr.

Given the increased noise in the Swath CryoSat-2 data compared to the point of closest approach product (POCA), this parameter was selected along with the others stated in this section to robustly identify and remove potential erroneous plane fits.

This value was selected as it encompassed the expected elevation changes over the AP region for our study period based on previous studies (Rott et al., 2018; Scambos et al., 2014), albeit we acknowledge some exceptional thinning at the Hektoria Green grounding line in the Northern Antarctic Peninsula during 2011–2013 may exceed this value (Rott et al., 2018). However, the inclusion of the complementary datasets such as the stereophotogrammetry data (which is more comprehensive in this northerly latitude compared to CryoSat-2), negates this issue somewhat as the signal would still be represented in the observations used in the BHM. The -10.1 m/yr value on page 15 refers only to the ice dynamics mass loss component – which is the dominant driver of mass losses in this region given the recent ice shelf collapse. We therefore believe that this choice of cut-off value is appropriate.

Pages 7–8, Lines 179–181: This is an important piece of the paper and should probably be in the abstract or conclusion.

Agreed. We have now added the following to the abstract to emphasise [L17]:

*"This is first time that such localised observations have been assimilated directly to estimate mass balance as part of a wider scale regional assessment"*

Page 8, Line 194: should be "photogrammetric" acquisitions correct?

Agreed. We have now changed as follows [L194-195]:

[L194-195] *"As the trends represent the multi-annual linear rate of change between two SAR stereo-photogrammetry acquisitions, we inflate the trend uncertainty using the approach of Eq.(1)."*

Page 8, Lines 208–211: A difficulty here would be the sensitivity of GRACE/GRACE-FO to near-grounding line change. If these values are masked in the DEM fields but of a large enough magnitude to be detectable by GRACE/GRACE-FO, then there has to be some sort of bias being introduced. Would be good to have some clarifying text in the discussion.

We understand the reviewers concern regarding this issue and have added the following to the "Limitations and Future Outlook" section at L534 (as we thought it was better suited here):

*"Additionally, it should be noted that the dDEM data pre-processing exclusion of elevation trends in regions of grounding line migration, may introduce a small bias with the GRACE data where large mass change signals occur in this vicinity. Whilst this bias is reduced compared to conventional altimetry techniques alone, it is still potentially present."*

We still believe that removing elevation trends on potentially ungrounded ice was the correct decision (as these elevation trends would not be due to an ice dynamics or SMB response we are solving for). However, we have added this clarification that there is a potential slight mismatch between the signals the different techniques are seeing. It should be noted that as GRACE is part of the observation suite, this signal is still seen in the framework, so any bias would be reduced compared to single technique approaches alone. We hope this statement clarifies this to the reader.

Page 9, Line 216: GIA is "quasi-invariant" over short time scales (years to decades). It would have been interesting to see the results if GIA wasn't fixed as the uncertainty is a limiting factor in GRACE/GRACE-FO studies (~4–5 Gt/yr difference between Ivins et al. (2013) and Whitehouse et al. (2012) in the AP)

We agree that an investigation into the quasi-invariant GIA solution would be interesting to undertake with the BHM. Solving simultaneously for a temporally evolving GIA solution is something that is currently being investigated for feasibility and would be useful to solve at an ice sheet scale given recent literature over the Amundsen Sea Embayment (Barletta et al., 2018) in addition to the AP. Given the extra methodological development and GPS data requirements we believe this could be a study in its own right and therefore was not explored fully in this paper. However, on Lines 550–556 of the manuscript we identify this as a potential future research avenue.

Page 11, Line 265: Is this 99.6% of the AP or ice sheet wide?

This is ice sheet wide and has been clarified in the text on line 265:

[Line 265] *"The 99.6% Antarctic Ice Sheet coverage in the updated product"*

Page 11, Lines 274–276: For overlapping grid cells in the 27km and 5.5km models, are the average spectral properties the same?

This is an analysis that was not undertaken as the 27km product was only used in regions that were outside the 5.5km model domain. The judgement was made that the higher resolution model was preferrable for use in regions where coverage is provided (as is the case for most studies in this region). It should be noted that the 27 km model is mainly used to provide coverage over the H-Hp drainage basin only (Abbot sector). Analysis between the two models has shown that low accumulation regions are equally well represented when compared against SMB observations (Van Wessem et al., 2016). However, at higher elevations in the AP, the 5.5km model better represents the patterns.

Page 11, Line 277: Use something besides "per se"?

We have replaced this phrasing with the following on Line 277:

[Line 277] *"by not using the raw model output"*

Page 11, Lines 287–289: The near collapse of Larsen-B resulted in rapid sub-annual responses in ice velocity (Rignot et al., 2004). Would this still be over-smoothing potential discharge contributions?

We acknowledge that this is a potential limitation on lines 522-524 of the manuscript (Limitations and Future Outlook), but an element of smoothness in the ice dynamic process is required for achieving source separation with SMB (a much more spatio-temporally variable process). Investigating sub-annual responses in ice dynamics would be an interesting application of the BHM but is currently limited by the temporal baseline of the elevation rates in satellite observations. ICESat-2 may make this more achievable in the future.

Page 14, Line 324: Figure 4 instead of "Fig. Figure 4"

This has been amended to the following in accordance with the TC manuscript guidelines

[Line 324] *"(Fig. Figure )"*

Page 16, Line 367: "Changes in surface processes are the major drivers"

We have now changed the text as per the reviewer's suggestion:

[Line 367] *"Changes in surface processes are the major drivers of inter-annual variability in mass balance,..."*

Page 16, Line 380: Could other altimetry datasets (OIB/IS2) help with the "paucity" of data?

ICEsat-2, since its launch in 2018, would reduce the paucity of the data, and we have acknowledged this in Line 535 of the 'future outlook section'. As the mission had only been in orbit for just over a year in the period of our study, it was at too early a stage to incorporate into this solution. However, it is something that is planned for incorporating when extending the approach into the next decade. See response to broad comments regarding the inclusion of OIB.

Page 16, Line 382: If it is a large enough signal in magnitude, GRACE/GRACE-FO should be able to observe it. However, it would be a smoothed representation of that mass change and difficult to localize due to the non-uniqueness of the solutions.

We acknowledge that GRACE does not suffer from paucity of coverage and have reworded Line 382 accordingly:

[Line 382] *"This is most likely due to the paucity of altimetry observations available in this region, despite the increase in observational coverage provided by the CryoSat-2 Swath data, inhibiting localisation of the smooth signal representation that would be shown by GRACE."*

Page 19, Lines 413–415: What about comparing to the individual components of IMBIE?

We agree it would be interesting to compare data from the IMBIE project on a per technique (GRACE, IOM, Altimetry) and a per process (SMB, Ice Dynamics) basis. However, as of the time of writing the manuscript, the data provided by the IMBIE project consisted of the cumulative mass balance time series for all three approaches combined (http://imbie.org/data-downloads/), so it is not possible to do a more in depth analysis. Partitioning of mass balance into component processes is a planned part of future IMBIE exercises and would be a welcome comparison to the BHM approach.

Page 21, Lines 467: "cannot be resolved with gravimetry alone."

Changed as per reviewer suggestion on Line 467:

[L467] *"which cannot be resolved by gravimetry alone"*

Page 23, Lines 486: Is there a space missing between "2011" and "to"?

Correct, this has been amended

Page 23, Lines 498–499: Would the deficiencies in calculating ice discharge in the Martín Español et al. (2016) solutions affect the GIA solution used here?

There is a chance that the iterative solution used to elastically correct the GPS trends, and then determine the GIA solution in the Martín Español et al. (2016) work, may affect the GIA solution here.

However, this solution was chosen due to its data-driven methodological approach and its use of GPS observations to constrain the posterior field, which was noted to contribute to its good

performance over the Antarctic Peninsula when compared against a variety of other GIA models (Martín-Español, King, et al., 2016). The use of any forward GIA model would also similarly be subject to biases and unaccounted for uncertainties and therefore subsequently impact the solution if used. As mentioned earlier in the responses, a future study investigating spatio-temporal variation in the GIA field using a BHM approach would be of interest.

References used in response

Agosta, C., Amory, C., Kittel, C., Orsi, A., Favier, V., Gallée, H., et al. (2019). Estimation of the Antarctic surface mass balance using the regional climate model MAR (1979-2015) and identification of dominant processes. *Cryosphere*, *13*(1), 281–296. https://doi.org/10.5194/tc-13-281-2019

Barletta, V. R., Bevis, M., Smith, B. E., Wilson, T., Brown, A., Bordoni, A., et al. (2018). Observed rapid bedrock uplift in amundsen sea embayment promotes ice-sheet stability. *Science*, *360*(6395), 1335–1339. https://doi.org/10.1126/science.aao1447

Depoorter, M. A., Bamber, J. L., Griggs, J., Lenaerts, J. T. M., Ligtenberg, S. R. M., van den Broeke, M. R., & Moholdt, G. (2013). Synthesized grounding line and ice shelf mask for Antarctica. *Supplement to: Depoorter, Mathieu A; Bamber, Jonathan L; Griggs, Jennifer; Lenaerts, Jan T M; Ligtenberg, Stefan R M; van Den Broeke, Michiel R; Moholdt, Geir (2013): Calving Fluxes and Basal Melt Rates of Antarctic Ice Shelves. Nature, 502, 89-92, Doi:10*. PANGAEA. https://doi.org/10.1594/PANGAEA.819151

Martín-Español, A., King, M. A., Zammit-Mangion, A., Andrews, S. B., Moore, P., & Bamber, J. L. (2016). An assessment of forward and inverse GIA solutions for Antarctica. *Journal of Geophysical Research: Solid Earth*, *121*(9), 6947–6965. https://doi.org/10.1002/2016JB013154

Martín-Español, A., Zammit-Mangion, A., Clarke, P. J., Flament, T., Helm, V., King, M. a, et al. (2016). Spatial and temporal Antarctic Ice Sheet mass trends, glacio-isostatic adjustment, and surface processes from a joint inversion of satellite altimeter, gravity, and GPS data. *Journal of Geophysical Research: Earth Surface*, *121*(2), 182–200. https://doi.org/10.1002/2015JF003550

McMillan, M., Shepherd, A., Sundal, A., Briggs, K., Muir, A., Ridout, A., et al. (2014). Increased ice losses from Antarctica detected by CryoSat-2. *Geophysical Research Letters*, *41*(11), 3899–3905. https://doi.org/10.1002/2014GL060111

Melchior Van Wessem, J., Jan Van De Berg, W., Noël, B. P. Y., Van Meijgaard, E., Amory, C., Birnbaum, G., et al. (2018). Modelling the climate and surface mass balance of polar ice sheets using RACMO2 - Part 2: Antarctica (1979-2016). *Cryosphere*, *12*(4), 1479–1498. https://doi.org/10.5194/tc-12-1479-2018

Rignot, E, Mouginot, J., & Scheuchl, B. (2011). Antarctic grounding line mapping from differential satellite radar interferometry. *Geophysical Research Letters*, *38*(10), n/a-n/a. https://doi.org/10.1029/2011GL047109

Rignot, E, Jacobs, S., Mouginot, J., & Scheuchl, B. (2013). Ice-Shelf Melting Around Antarctica. *Science*, *341*(6143), 266–270. https://doi.org/10.1126/science.1235798

Rignot, Eric, Mouginot, J., Scheuchl, B., van den Broeke, M., van Wessem, M. J., & Morlighem, M. (2019). Four decades of Antarctic Ice Sheet mass balance from 1979–2017. *Proceedings of the National Academy of Sciences*, 201812883. https://doi.org/10.1073/pnas.1812883116

Rott, H., Abdel Jaber, W., Wuite, J., Scheiblauer, S., Floricioiu, D., van Wessem, J. M., et al. (2018). Changing pattern of ice flow and mass balance for glaciers discharging into the Larsen A and B embayments, Antarctic Peninsula, 2011 to 2016. *The Cryosphere*, *12*(4), 1273–1291. https://doi.org/10.5194/tc-12-1273-2018

Scambos, T. A., Berthier, E., Haran, T., Shuman, C. A., Cook, A. J., Ligtenberg, S. R. M., & Bohlander, J. (2014). Detailed ice loss pattern in the northern Antarctic Peninsula: widespread decline driven by ice front retreats. *The Cryosphere*, *8*(6), 2135–2145. https://doi.org/10.5194/tc-8-2135-2014

Schoen, N., Zammit-Mangion, A., Rougier, J. C., Flament, T., Rémy, F., Luthcke, S., & Bamber, J. L. (2015). Simultaneous solution for mass trends on the West Antarctic Ice Sheet. *The Cryosphere*, *9*(2), 805–819. https://doi.org/10.5194/tc-9-805-2015

Shepherd, A., Ivins, E., Rignot, E., Smith, B., van den Broeke, M., Velicogna, I., et al. (2018). Mass balance of the Antarctic Ice Sheet from 1992 to 2017. *Nature*, *558*(7709), 219–222. https://doi.org/10.1038/s41586-018-0179-y

Shepherd, A., Gilbert, L., Muir, A. S., Konrad, H., McMillan, M., Slater, T., et al. (2019). Trends in Antarctic Ice Sheet Elevation and Mass. *Geophysical Research Letters*, 2019GL082182. https://doi.org/10.1029/2019GL082182

Van Wessem, J. M., Ligtenberg, S. R. M., Reijmer, C. H., Van De Berg, W. J., Van Den Broeke, M. R., Barrand, N. E., et al. (2016). The modelled surface mass balance of the Antarctic Peninsula at 5.5⁻km horizontal resolution. *Cryosphere*, *10*(1), 271–285. https://doi.org/10.5194/tc-10-271-2016

Zammit-Mangion, A., Rougier, J., Schön, N., Lindgren, F., & Bamber, J. (2015). Multivariate spatio-temporal modelling for assessing Antarctica's present-day contribution to sea-level rise. *Environmetrics*, *26*(3), 159–177. https://doi.org/10.1002/env.2323